# Pareto-Optimal Learning-Augmented Algorithms for Online Conversion Problems

**Bo Sun**
ECE, HKUST
bsunaa@connect.ust.hk

**Russell Lee**
CICS, UMass Amherst
rclee@cs.umass.edu

**Mohammad Hajiesmaili**
CICS, UMass Amherst
hajiesmaili@cs.umass.edu

**Adam Wierman**
CMS, Caltech
adamw@caltech.edu

**Danny H.K. Tsang**
ECE, HKUST
eetsang@ust.hk

## Abstract

This paper leverages machine-learned predictions to design competitive algorithms for online conversion problems with the goal of improving the competitive ratio when predictions are accurate (i.e., consistency), while also guaranteeing a worst-case competitive ratio regardless of the prediction quality (i.e., robustness). We unify the algorithmic design of both integral and fractional conversion problems, which are also known as the 1-max-search and one-way trading problems, into a class of online threshold-based algorithms (OTA). By incorporating predictions into design of OTA, we achieve the Pareto-optimal trade-off of consistency and robustness, i.e., no online algorithm can achieve a better consistency guarantee given for a robustness guarantee. We demonstrate the performance of OTA using numerical experiments on Bitcoin conversion.

## 1 Introductions

An online conversion problem aims to convert one asset to another through a sequence of exchanges at varying rates in order to maximize the terminal wealth in financial markets. With limited information on possible future rates, the core challenge in an online conversion problem is how to balance the return from waiting for possible high rates with the risk that high rates never show up. A high profile example of this risk is cryptocurrency markets, e.g., Bitcoin, where high fluctuations up and down make it challenging to optimize exchanges. Two well-known classical online conversion problems are 1-max-search [7] and one-way trading [8], which can be considered as integral and fractional versions of the online conversion problem that trade the asset as a whole or fraction-by-fraction (e.g., trading stock in lot or shares). Beyond these two problems, a number of extensions and variants of online conversion problems have been studied with applications to lookback options [14], online portfolio selection [13], online bidding [5], and beyond.

Most typically, conversion problems are studied through the lens of competitive ratios and the goal is to design online algorithms that minimize the worst-case return ratio of the offline optimal to online algorithm decisions. For example, EI-Yaniv et al. [8] have shown that optimal online algorithms can be designed to achieve the minimal competitive ratios for both 1-max-search and one-way trading. However, in real-world problems, predictions about future conversion rates are increasingly available and the algorithms developed in the literature are not designed to take advantage of such information. The challenge for using such predictions is that, in one extreme, the additional information is an accurate prediction (advice) of future inputs. In this case, the algorithm can confidently use the information to improve performance, e.g., [9]. However, most commonly, predictions have

35th Conference on Neural Information Processing Systems (NeurIPS 2021).

no guarantees on their accuracy, and if an online algorithm relies on an inaccurate prediction the performance can be even worse than if it had ignored the prediction entirely.

This challenge is driving the emerging area of the learning-augmented online algorithm (LOA) design, which seeks to design online algorithms that can incorporate untrusted machine-learned predictions in a way that leads to near-optimal performance when predictions are accurate but maintains robust performance when predictions are inaccurate. To measure this trade-off, two metrics have emerged, introduced by [15] and [19]: *consistency* and *robustness*. Consistency is defined as the competitive ratio when the prediction is accurate, i.e., $\mathtt{CR}(0)$, where $\mathtt{CR}(\varepsilon)$ is the competitive ratio when the prediction error is $\varepsilon$. In contrast, *robustness* is the worst competitive ratio over any prediction errors, i.e., $\max_\varepsilon \mathtt{CR}(\varepsilon)$. Thus, consistency and robustness provide a way to quantify the ability of an algorithm to exploit accurate predictions while ensuring robustness to poor predictions.

In recent years, a stream of literature has sought to design robust and consistent LOA for a variety of online problems, such as online caching [15], ski-rental [19], and others. The ultimate goal is to develop algorithms that are *Pareto-optimal* across robustness and consistency, in the sense that for any $\gamma$, the LOA achieves the minimal consistency guarantee among all online algorithms that are $\gamma$-competitive. For the ski rental problem, recent works have derived Pareto-optimal algorithms, e.g., [4, 22], but in most cases the question of where the Pareto-boundary of LOA lies is yet to be answered.

In this paper, we focus on the design of LOA for online conversion problems and we seek to answer the following question: *Is it possible to design a Pareto-optimal* LOA *for the online conversion problem?*

**Contributions.** We show that the answer to the above question is "yes", by designing an online threshold-based algorithm (OTA), and proving that it is Pareto-optimal. In particular, we introduce a class of OTA that unifies the algorithmic design of both 1-max-search and one-way trading. We then incorporate predictions into OTA by parameterizing the threshold functions based on the predictions. This approach yields bounded consistency and robustness (see Theorem 4.5 and Theorem 4.6). Further, we derive lower bounds for robustness-consistency trade-offs and show that our learning-augmented OTA achieves those lower bounds, and is thus Pareto-optimal (see Theorem 5.1 and Theorem 5.2). Finally, we demonstrate the improvement of the learning-augmented OTA over pure online algorithms using numerical experiments based on real-world data tracking Bitcoin prices.

The technical contributions of this paper are twofold. First, we provide a sufficient condition for design and analysis of the learning-augmented OTA with a guaranteed generalized competitive ratio. This competitive ratio is general in the sense that it not only can yield robustness and consistency guarantees, but can also potentially provide more fine-grained performance guarantees beyond robustness and consistency. Second, we provide a novel way of deriving the lower bound on the robustness-consistency trade-off, which may be of use beyond online conversion problems. The key idea is to construct a function that can model all online algorithms under a special family of instances, and the lower bound can be derived from combining the robustness and consistency requirements on this function. This constructive approach to arriving at a lower bound is distinctive.

## 2 Related Work

**Online Conversion Problems.** The online conversion problem is first introduced by El-Yaniv [7] and analyzed under the competitive analysis framework. It aims to search for the maximum rate to make conversions from a sequence of time-varying rates, which are chosen by an adversary in the worst-case. The subsequent works [8, 14] have designed competitive algorithms for both integral and continuous versions of the online conversion problem and achieved the minimal competitive ratios. Many variants and extensions have also been studied, such as online conversion under interrelated rates [20] or with inventory constraints [23, 12]. See [18] for a survey of other variants. In addition, the design and analysis of competitive algorithms for the online conversion problem have been shown to closely relate to those of the online knapsack problems, such as the online $0/1$ knapsack with small weights [24, 26] and online fractional knapsack [21]. Beyond the worst-case input model, another line of research considers searching for the maximum value from the time-varying rates with some form of prior knowledge. For example, the secretary problem [3] assumes that the rates arrive in a uniformly random order and the prophet inequality [6] assumes that the rates arrive in the worst-case order but with prior distributions. The prior information in these problems makes the design and analysis of algorithms essentially different from those of online conversion problems.

**Learning-augmented algorithms.** The learning-augmented algorithm (LOA) takes advantages of machine learned predictions about the future input into the design of online algorithms and aims to optimize for a better competitive performance with an improved quality of predictions. The concepts of consistency and robustness are first introduced to study the online caching problem in [15]. Then the follow-up work [19] formally shows that there exist trade-offs between consistency and robustness in the ski-rental problem and its variants, and LOA needs to be designed to balance the two criteria unlike the online algorithms just optimized for the competitive ratio. Furthermore, the works [4] and [22] independently prove that the LOA designed in [19] is Pareto-optimal, i.e., given a robustness, no online algorithms can achieve a smaller consistency. Also, in [11] and motivated by an energy optimization scenario, the authors develop Pareto-optimal algorithms for an extended version of the ski-rental problem. From the aspect of methodology, the work [4] provides a general approach to incorporate the prediction into the online primal-dual algorithm and shows it can be applied to solve multiple online problems. In terms of applications, LOA has been designed for a variety of online problems, such as online caching [15], ski-rental [19, 22, 4, 1, 11], online set cover [4], online scheduling [19, 10], secretary and online matching [3], metrical task systems [2], etc. However, among those applications, the Pareto-optimality of LOA has only been rigorously shown for the ski-rental problem.

## 3   Problem statement and a unified algorithm

**The online conversion problem.** An online conversion problem considers how to convert one asset (e.g., dollars) to another (e.g., yens) over a trading period $[N] := \{1, \ldots, N\}$. At the beginning of step $n \in [N]$, an exchange rate (or price), $v_n$, is announced and a decision maker must immediately determine the amount of dollars, $x_n$, to convert and obtains $v_n x_n$ yens. The conversion is unidirectional, i.e., yens are not allowed to convert back to dollars. The trading horizon $N$ is unknown to the decision maker, and if there are any remaining dollars after $N-1$ trading steps, all of them will be compulsorily converted to yens at the last price $v_N$. Without loss of generality, the initial asset can be assumed to be 1 dollar, and the goal is to maximize the amount of yens acquired at the end of the trading period. The offline version of the conversion problem can be cast as

$$\underset{x_n}{\text{maximize}} \quad \sum_{n \in [N]} v_n x_n, \quad \text{subject to} \quad \sum_{n \in [N]} x_n \leq 1. \tag{1}$$

If the conversion is only allowed in a single transaction, the decision $x_n \in \{0, 1\}$ is a binary variable, and this integral version is called 1-*max-search* [7]. If the asset is allowed to convert fraction-by-fraction over multiple transactions, the decision $x_n \in [0, 1]$ is a continuous variable, and this fractional version is refereed to as *one-way trading* [8]. Following the literature, we assume the prices $\{v_n\}_{n \in [N]}$ are bounded, i.e., $v_n \in [L, U], \forall n \in [N]$, where $L$ and $U$ are known parameters, and define $\theta = U/L$ as the price fluctuation.

**Online threshold-based algorithms.** Online threshold-based algorithms (OTA) belong to a class of *reserve-and-greedy* algorithms where the idea is to use a threshold function to determine the amount of resources that need to be *reserved* based on resource utilization, and then *greedily* allocate resources respecting the reservation in each step.

OTA is known to be easy-to-use but hard-to-design due to the difficulties in developing the threshold function. Prior work using OTA has often been problem specific. For example, an optimal design of the threshold function in OTA is derived in [21] for one-way trading and in [25, 26] for the online knapsack problem, which is closely related to one-way trading.

Here, we unify online algorithms for online conversion problems in an OTA framework in Algorithm 1. Algorithm 1 takes a threshold function $\phi$ as its input, where $\phi(w) : [0, 1] \to [L, U]$ is a function of resource utilization (i.e., amount of traded dollar) $w$ and $\phi(w)$ can be considered as the reservation price when the utilization is $w$. The algorithm makes conversions only if the current price $v_n$ is at least $\phi(w^{(n-1)})$, where $w^{(n-1)} = \sum_{i \in [n-1]} \bar{x}_i$ is the utilization after the previous $n-1$ steps of trading. More specifically, the conversion decision $\bar{x}_n$ in each step is determined by solving an optimization problem in Line 3 of Algorithm 1. The OTA framework transforms the algorithmic design task in online conversion problems into the design of $\phi$, and the challenge is to design $\phi$ such that OTA can have theoretical performance guarantees. To provide two examples, in the following we show how to recover the optimal online algorithms for 1-max-search and one-way trading.

---
**Algorithm 1** Online threshold-based algorithm with threshold function $\phi$ ($\mathtt{OTA}_\phi$)
---
1: **input:** threshold function $\phi(\cdot)$, and initial resource utilization (i.e., traded dollar) $w^{(0)} = 0$;
2: **while** price $v_n$ is revealed **do**
3:     determine resource allocation $\bar{x}_n = \arg\max_{x_n \in \mathcal{X}_n} v_n x_n - \int_{w^{(n-1)}}^{w^{(n-1)}+x_n} \phi(u)du$;
4:     update the utilization $w^{(n)} = w^{(n-1)} + \bar{x}_n$.
5: **end while**
---

*1-max-search.* In this integral conversion problem, $\mathtt{OTA}$ sets the feasible space as $\mathcal{X}_n = \{0,1\}$ and the threshold function as a constant $\phi(w) = \Phi, w \in [0,1]$, where $\Phi$ is also called a reservation price. Then the algorithm simply selects the first price that is at least $\Phi$. When the reservation price is designed as $\Phi = \sqrt{LU}$, $\mathtt{OTA}$ is exactly the same algorithm as the reservation price policy in [8], which achieves the optimal competitive ratio $\sqrt{\theta}$.

*One-way trading.* $\mathtt{OTA}$ sets $\mathcal{X}_n = [0, 1 - w^{(n-1)}]$ and $\phi$ as a continuous and strictly increasing function. The conversion decisions fall into three cases based on the solution of the optimization in Line 3: (i) if $v_n < \phi(w^{(n-1)})$, make no conversions, i.e., $\bar{x}_n = 0$; (ii) if $\phi(w^{(n-1)}) \le v_n \le \phi(1)$, $\bar{x}_n$ can be solved based on the first-order optimality condition, i.e., $v_n = \phi(w^{(n-1)} + \bar{x}_n)$; and (iii) if $v_n > \phi(1)$, $\bar{x}_n = 1 - w^{(n-1)}$ converts all its remaining dollar at the price $v_n$. By setting the threshold function to $\phi(w) = L + (\alpha^* L - L) \exp(\alpha^* w), w \in [0,1]$, $\mathtt{OTA}$ achieves the optimal competitive ratio $\alpha^* = 1 + W((\theta - 1)/e)$, where $W(\cdot)$ is the Lambert-W function [21].

## 4   Robustness and consistency

This paper is focused on the design of learning-augmented online algorithms ($\mathtt{LOA}$) where the online algorithm is given a machine-learned prediction $P \in [L, U]$ of the maximum price $V = \max_{n \in [N]} v_n$ over the price sequence. In the online conversion problem, suppose $V$ is known a prior, waiting to trade all assets at the price $V$ achieves the maximum profit. Therefore, the maximum price is an appropriate value to be predicted and incorporated into the design of the online algorithm.   The prediction $P$ is not necessarily accurate and we define $\varepsilon = |V - P|$ as the prediction error. Let $\mathtt{CR}(\varepsilon)$ denote the competitive ratio of $\mathtt{OTA}$ when the prediction error is $\varepsilon$. Our goal is to design an algorithm that is $\eta$-consistent and $\gamma$-robust, i.e., an algorithm where $\eta \ge \mathtt{CR}(0)$ and $\gamma \ge \max_\varepsilon \mathtt{CR}(\varepsilon)$. We first focus on designing a learning-augmented $\mathtt{OTA}$ by incorporating predictions into the design of the threshold function $\phi$ to achieve bounded robustness and consistency.

### 4.1   Warmup

To highlight the challenges of algorithm design in this setting, we start by showing that an intuitive use of predictions can result in poor robustness-consistency guarantees. Thus, it is of essential importance to take advantage of the problem structure in designing the learning-augmented $\mathtt{OTA}$.

To illustrate this, we consider the design of the reservation price $\Phi_P$ for 1-max-search as an example. If we blindly use the prediction of the maximum price by setting $\Phi_P = P$, $\mathtt{OTA}$ is indeed offline optimal when the prediction is accurate, and thus 1-consistent. However, its robustness is the worst possible competitive ratio $\theta$, which is achieved when the prediction is $P = U$ and the actual maximum price is $V = U - \epsilon$, where $\epsilon \to 0$. In fact, the robustness guarantee approaches $\theta$ with an arbitrarily small prediction error.

Another intuitive design is to set the reservation price as a linear combination of $P$ and the optimal reservation price for pure online algorithms $\sqrt{LU}$, i.e., $\Phi_P = \lambda\sqrt{LU} + (1-\lambda)P$, where $\lambda \in [0,1]$ is called the *robustness parameter*, indicating the distrust in the prediction. The robustness and consistency of this algorithm is characterized by the following result.

**Proposition 4.1.** *Given $\lambda \in (0,1]$, $\mathtt{OTA}$ with the reservation price $\Phi_P = \lambda\sqrt{LU} + (1-\lambda)P$ for 1-max-search is $(\lambda\sqrt{\theta} + (1-\lambda)\theta)$-robust and $\sqrt{\theta}$-consistent.*

Above consistency-robustness result is tight in the sense that for a given $\lambda$, we can construct an instance, under which $\mathtt{OTA}$ with the reservation price $\Phi_P$ can achieve the robustness and consistency

in Proposition 4.1 with arbitrarily small gaps. This result highlights that, while the robustness is a linear combination of the optimal competitive ratio $\sqrt{\theta}$ and $\theta$, the consistency is $\sqrt{\theta}$, which yields no improvement over the optimal competitive ratio except for a special case when $\lambda = 0$.

## 4.2 A sufficient condition

Together, the two examples in the previous section highlight some of the challenges associated with balancing robustness and consistency in OTA. Given the challenges, we now focus on developing a general approach for the design and analysis of robust and consistent learning-augmented OTA. To do so, we first generalize the competitive ratio from a scalar to a vector, where each element corresponds to a competitive ratio over a subset of instances (see Definition 4.2). Since consistency and robustness can be considered as the competitive ratios over the subsets of the predicted instances and the other instances, the competitiveness of OTA can be transformed to robustness-consistency guarantees (see Lemma 4.3). This transformation leads us to characterize a general sufficient condition (see Theorem 4.4) on the threshold function of OTA that guarantees a generalized competitive ratio over a given subsets of instances. Then, combining Theorem 4.4 and Lemma 4.3 gives a general approach for analyzing the consistency and robustness of OTA, which we leverage in the analysis of 1-max-search and one-way trading in the following sections in order to illustrate its applicability.

To begin, let $\mathrm{OPT}(\mathcal{I})$ and $\mathrm{ALG}(\mathcal{I})$ denote the returns of offline optimal and an online algorithm under instance $\mathcal{I}$, respectively. Let $\mathcal{P} := \{\mathcal{P}_1, \ldots, \mathcal{P}_I\}$ be a partition of the set $\Omega$ of all instances.

**Definition 4.2** (Generalized competitive ratio). $\boldsymbol{\alpha} := (\alpha_1, \ldots, \alpha_I)$ *is a generalized competitive ratio over* $\mathcal{P}$ *if* $\alpha_i = \max_{\mathcal{I} \in \mathcal{P}_i} \mathrm{OPT}(\mathcal{I})/\mathrm{ALG}(\mathcal{I})$ *is the worst-case ratio over* $\mathcal{P}_i$ *for all* $i \in [I]$.

In online conversion problems, let $\Omega_p \subseteq \Omega$ be a subset, in which each instance has a maximum price $p$. Thus, if we have a prediction $P$ on the maximum price, it means the instance is predicted to belong to $\Omega_P$. We can show the consistency and robustness of a learning-augmented OTA by proving its generalized competitive ratio over a partition. In particular, given prediction $P$, OTA is $\eta$-consistent and $\gamma$-robust if there exists a partition such that OTA is $\eta$-competitive over the subset that contains $\Omega_P$ and $\gamma$-competitive for the remaining subsets. Formally we have the following claim.

**Lemma 4.3.** *Given a prediction* $P \in [L, U]$, *and parameters* $\eta$ *and* $\gamma$ *with* $\eta \le \gamma$, OTA *for online conversion problems is* $\eta$-consistent *and* $\gamma$-robust *if there exists a partition* $\mathcal{P} = \{\mathcal{P}_\eta, \mathcal{P}_\gamma\}$ *with* $\Omega = \mathcal{P}_\eta \cup \mathcal{P}_\gamma$ *and* $\Omega_P \subseteq \mathcal{P}_\eta$, *and* OTA *is* $(\eta, \gamma)$-competitive *over* $\mathcal{P}$.

Building on Lemma 4.3, we now focus on how to design the threshold function $\phi$ in OTA to ensure a small generalized competitive ratio. To this end, divide the range of price $[L, U]$ into $I$ price segments $[M_0, M_1), \ldots, [M_{I-1}, M_I]$ with $L = M_0 < M_1 < \ldots < M_I = U$. We partition $\Omega$ based on the price segments, i.e., $\Omega = \{\Omega_p\}_{p \in [M_0, M_1)} \cup \cdots \cup \{\Omega_p\}_{p \in [M_{I-1}, M_I]}$. Hereafter, we use $\mathcal{P}_i$ or $[M_{i-1}, M_i)$ to denote the $i$-th instance subset $\{\Omega_p\}_{p \in [M_{i-1}, M_i)}$. To ensure different worst-case ratios over different subsets of instances, we consider a piece-wise threshold function $\phi$ created by concatenating a sequence of functions $\{\phi_i\}_{i \in [I]}$, where each piece $\phi_i$ is designed to guarantee $\alpha_i$-competitiveness over $\mathcal{P}_i$. In particular, divide the feasible region $[0, 1]$ into $I$ resource segments $[\beta_0, \beta_1), \ldots, [\beta_{I-1}, \beta_I]$ with $0 = \beta_0 \le \beta_1 \le \cdots \le \beta_I = 1$, and $\phi_i(w) \in [M_{i-1}, M_i), w \in [\beta_{i-1}, \beta_i)$. We say $\phi_i$ is absorbed if $\beta_{i-1} = \beta_i$. The following theorem then provides a sufficient condition for designing the threshold function $\phi$ in OTA to guarantee a generalized competitive ratio.

**Theorem 4.4.** OTA *is* $\boldsymbol{\alpha}$-competitive *over* $\{\mathcal{P}_i\}_{i \in [I]}$ *for online conversion problems if* $\phi := \{\phi_i\}_{i \in [I]}$ *is a piece-wise and right-continuous function,* $\phi(1) \in \{M_i\}_{i \in [I]}$ *is one of the partition boundaries, and each threshold piece* $\phi_i(w) : [\beta_{i-1}, \beta_i) \to [M_{i-1}, M_i)$ *satisfies one of the following conditions:*

Case I: *if* $M_i \le \phi(0)$, *then* $M_i \le \alpha_i L$ *and* $\beta_i = 0$;

Case II: *if* $\phi(0) < M_i \le \phi(1)$, *then* $\phi_i$ *is in the form of*

$$\phi_i(w) = \begin{cases} M_{i-1} & w \in [\beta_{i-1}, \beta'_{i-1}) \\ \varphi_i(w) & w \in [\beta'_{i-1}, \beta_i) \end{cases}, \tag{2}$$

*which consists of a flat segment in* $[\beta_{i-1}, \beta'_{i-1})$ *and a strictly increasing segment* $\varphi_i(w)$ *that satisfies*

$$\begin{cases} \varphi_i(w) \le \alpha_i \left[ \int_0^{\beta'_{i-1}} \phi(u) du + \int_{\beta'_{i-1}}^w \varphi_i(u) du + (1-w)L \right], \forall w \in [\beta'_{i-1}, \beta_i) \\ \varphi_i(\beta_i) = M_i \end{cases} ; \tag{3}$$

Case III: *if $M_i > \phi(1)$, then $M_i \leq \alpha_i \int_0^1 \phi(u)du$ and $\beta_i = 1$.*

Theorem 4.4 is the key to the analysis that follows. In particular, it provides a sufficient condition for analyzing the generalized competitive ratio of `OTA`, which in turn yields bounds on consistency and robustness as highlighted in Lemma 4.3. We show its broad applicability in the subsections that follow by applying it in the context of 1-max search and one-way trading.

Further, this approach is general and provides opportunities to derive more fine-grained performance metrics for the learning-augmented `OTA` beyond consistency and robustness. For example, instead of just focusing on the improved competitive ratio when predictions are accurate, we can redefine the consistency as a prediction-error dependent metric $\kappa(\xi) := \max_{\varepsilon \leq \xi} \text{CR}(\varepsilon)$. $\kappa(\xi)$ characterizes the improved competitive ratio if the actual value is within the neighbourhood of the prediction $[P - \xi, P + \xi]$. Thus, $\kappa(\xi)$ is a more general and fine-grained metric, and $\eta$ and $\gamma$ are two extreme points of $\kappa(\xi)$, i.e., $\eta = \kappa(0)$ and $\gamma = \kappa(\infty)$. Given $\xi$, we can leverage the competitive ratio to design $\kappa(\xi)$-consistent and $\gamma$-robust `OTA`. In particular, given $P$, `OTA` is $\kappa(\xi)$-consistent and $\gamma$-robust if there exists a partition $\mathcal{P} = \{\mathcal{P}_{\kappa(\xi)}, \mathcal{P}_\gamma\}$ with $\Omega_{p \in [P-\xi, P+\xi]} \subseteq \mathcal{P}_{\kappa(\xi)}$, and `OTA` is $(\kappa(\xi), \gamma)$-competitive over $\mathcal{P}$.

### 4.3   1-max search

We now apply the sufficient condition in Theorem 4.4 to the setting of 1-max search. Our goal is to design the reservation price $\Phi_P$ given a prediction $P$. To do this, we set $\eta := \eta(\lambda)$ and $\gamma := \gamma(\lambda)$ as

$$\gamma(\lambda) = [\sqrt{(1-\lambda)^2 + 4\lambda\theta} - (1-\lambda)]/(2\lambda), \text{ and } \eta(\lambda) = \theta/\gamma(\lambda), \tag{4}$$

where $\lambda \in [0, 1]$ is the robustness parameter, and $\eta$ and $\gamma$ are predetermined parameters for designing $\Phi_P$ that represent the consistency and robustness that we target to achieve. In particular, $\eta$ and $\gamma$ are designed as the solution of

$$\eta(\lambda) = \theta/\gamma(\lambda), \text{ and } \eta(\lambda) = \lambda\gamma(\lambda) + 1 - \lambda. \tag{5}$$

The first equation is the desired trade-off between robustness and consistency, which will be shown to match the lower bound in Section 5, and thus represents a Pareto-optimal trade-off. The second equation sets $\eta$ as a linear combination of 1 and $\gamma$. In this way, as $\lambda$ increases from 0 to 1 , $\eta$ increases from the best possible ratio 1 to the optimal competitive ratio $\sqrt{\theta}$, and $\gamma$ decreases from the worst possible ratio $\theta$ to $\sqrt{\theta}$. Taking $\eta$ and $\gamma$ as inputs, we design the reservation price $\Phi_P$ as follows:

$$\text{when } P \in [L, L\eta), \ \Phi_P = L\eta; \tag{6a}$$
$$\text{when } P \in [L\eta, L\gamma), \ \Phi_P = \lambda L\gamma + (1-\lambda)P/\eta; \tag{6b}$$
$$\text{when } P \in [L\gamma, U], \ \Phi_P = L\gamma. \tag{6c}$$

The following theorem provides robustness and consistency bounds for this algorithm. The result follows from the general sufficient condition for the class of `OTA` in Section 4.2. Given each reservation price $\Phi_P$, the key step of analysis is to determine a proper partition of instances and then analyze the competitive ratio over each subset, in which $\Phi_P$ can satisfy the sufficient condition in one of the cases in Theorem 4.4. Take $\Phi_P$ in (6a) for an example. We partition $[L, U]$ into $[L, L\eta)$ and $[L\eta, U]$ by letting $M_1 = L\eta$. Since $\phi(0) = \phi(1) = \Phi_P = L\eta$, $\Phi_P$ satisfies Case I and Case III for $[L, L\eta)$ and $[L\eta, U]$, respectively, and the corresponding competitive ratios are $\alpha_1 = \Phi_P/L = \eta$ and $\alpha_2 = U/\Phi_P = \theta/\eta = \gamma$. Thus, `OTA` is $(\eta, \gamma)$-competitive over $[L, L\eta)$ and $[L\eta, U]$. Additionally, the predicted instance $\Omega_P \subseteq \Omega_{p \in [L, L\eta)}$, and thus `OTA` is $\eta$-consistent and $\gamma$-robust based on Lemma 4.3.

**Theorem 4.5.** *Given $\lambda \in [0, 1]$, `OTA` with the reservation price in Equation (6) for 1-max-search is $\gamma(\lambda)$-robust and $\eta(\lambda)$-consistent, where $\gamma(\lambda)$ and $\eta(\lambda)$ are given in Equation (4).*

Before moving to the proof it is important to give insights into the form of the reservation price (6). It consists of three segments for predictions that are in boundary regions $[L, L\eta)$ and $[L\gamma, U]$ close to price lower and upper bounds, and in intermediate region $[L\eta, L\gamma)$. Figure 1 illustrates the form and compares it with two intuitive designs $\Phi_P = P$ and $\Phi_P = \lambda\sqrt{LU} + (1-\lambda)P$, which we have shown providing poor robustness and consistency guarantees. Given any reservation price $\Phi \in [L, U]$, the robustness of `OTA` is $\max\{\Phi/L, U/\Phi\}$, where $\Phi/L$ and $U/\Phi$ are the worst-case ratios over subsets $[L, \Phi)$ and $[\Phi, U]$. To ensure a good robustness, (6a) and (6c) are designed to balance $\Phi/L$ and $U/\Phi$

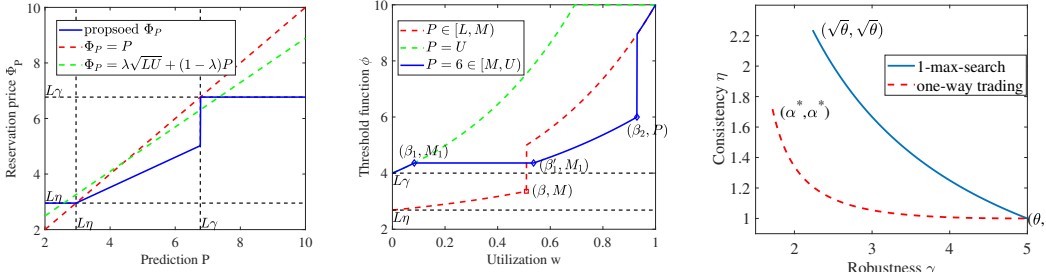

Figure 1: Reservation price and threshold function for 1-max-search (left) and one-way trading (right) with $L = 2$ and $U = 10$

Figure 2: Optimal robustness-consistency trade-offs

by just ensuring $\eta$-competitiveness over the boundary region that contains the prediction. The intuitive design $\Phi_P = P$ neglects this structure, and thus its robustness approaches the worst possible ratio $\theta$. Given an accurate prediction $P$, the consistency of OTA is $\max\{\Phi/L, P/\Phi\}$. To guarantee a good consistency, we must avoid the case that $P < \Phi$, leading to the ratio $\Phi/L$ that cannot be properly bounded. (6b) is designed by enforcing $P \geq \Phi_P$. In this way, OTA always makes conversions in the intermediate region and the consistency is $P/\Phi_P$, which can be designed to be upper bounded by $\eta$. The intuitive design $\Phi_P = \lambda\sqrt{LU} + (1 - \lambda)P$ fails to improve the consistency over $\sqrt{\theta}$ since it cannot always guarantee $P \geq \Phi_P$ in the intermediate region, and thus may make no conversions even with an accurate prediction. A full proof of Theorem 4.5 is in Appendix A.4.

### 4.4  One-way trading

Next, we apply the sufficient condition in Theorem 4.4 to one-way trading. We also aim to design the threshold function based on the prediction $P$. Here, we set $\gamma := \gamma(\lambda)$ and $\eta := \eta(\lambda)$ as

$$\eta(\lambda) = \theta / \left[ \frac{\theta}{\gamma(\lambda)} + (\theta - 1)\left(1 - \frac{1}{\gamma(\lambda)}\ln\frac{\theta - 1}{\gamma(\lambda) - 1}\right)\right], \text{ and } \gamma(\lambda) = \alpha^* + (1 - \lambda)(\theta - \alpha^*), \quad (7)$$

where $\lambda \in [0, 1]$ is the robustness parameter and $\alpha^*$ is the optimal competitive ratio of one-way trading. Similarly to the design in 1-max-search, the two equations in (7) determine the desired trade-off between $\eta$ and $\gamma$, and their desired relationship with $\lambda$. Again, we derive a lower bound in Section 5 showing that this relationship is tight and provides a Pareto-optimal trade-off.

Taking $\eta$ and $\gamma$ as inputs, we design the threshold function as follows:

$$\text{when } P \in [L, M), \ \phi_P(w) = \begin{cases} L + (\eta L - L)\exp(\eta w) & w \in [0, \beta) \\ L + (U - L)\exp(\gamma(w - 1)) & w \in [\beta, 1] \end{cases}, \quad (8a)$$

$$\text{when } P \in [M, U], \ \phi_P(w) = \begin{cases} L + (\gamma L - L)\exp(\gamma w) & w \in [0, \beta_1) \\ M_1 & w \in [\beta_1, \beta_1') \\ L + (M_1 - L)\exp(\eta(w - \beta_1')) & w \in [\beta_1', \beta_2] \\ L + (U - L)\exp(\gamma(w - 1)) & w \in (\beta_2, 1] \end{cases}, \quad (8b)$$

where $\beta$ and $M$ are solutions of

$$\begin{cases} M = L + (\eta L - L)\exp(\eta\beta), \\ M\gamma/\eta = L + (U - L)\exp(\gamma(\beta - 1)); \end{cases} \quad (9)$$

and $M_1, \beta_1, \beta_1'$, and $\beta_2$ are all functions of $P$ and are determined by

$$\begin{cases} \beta_1 = \frac{1}{\gamma}\ln\frac{\max\{M_1/L, \gamma\} - 1}{\gamma - 1}, \\ \frac{M_1}{\eta} = \int_0^{\beta_1}\phi(u)du + (\beta_1' - \beta_1)M_1 + (1 - \beta_1')L, \\ P = L + (M_1 - L)\exp(\eta(\beta_2 - \beta_1')), \\ \beta_2 = 1 + \frac{1}{\gamma}\ln\frac{\min\{P\gamma/\eta, U\} - L}{U - L}. \end{cases} \quad (10)$$

The following theorem provides robustness and consistency bounds for this algorithm. Again, the result follows from the general sufficient condition for the class of OTA that we introduce in Section 4.2.

Compared to 1-max-search, the additional difficulty of one-way trading lies in the analysis of the competitive ratios over the subsets belonging to Case II of the sufficient condition since the threshold function $\phi_P$ ranging in these subsets needs to satisfy a set of differential equations (3). $\phi_P$ in (8) is in fact designed as the solution of the differential equation (3) with binding inequalities and properly designed boundary conditions (by setting the length of the flat segment of each threshold piece).

**Theorem 4.6.** *Given $\lambda \in [0,1]$,* OTA *with the threshold function* (8) *for one-way trading is $\gamma(\lambda)$-robust and $\eta(\lambda)$-consistent, where $\gamma(\lambda)$ and $\eta(\lambda)$ are given in Equation* (7).

Figure 1 illustrates the function given different predictions. The basic idea behind the design of the threshold function (8) is similar to that of 1-max-search. When the prediction is in the boundary region $P \in [L, M)$, the threshold function (8a) (i.e., red curve) is designed to ensure $\eta$-competitiveness over $[L, M]$, and additionally guarantee $\gamma$-competitiveness over $[M, U]$. In the other extreme when $P = U$, the threshold function (8b) becomes $\phi_P(w) = L + (L\gamma - L)\exp(\gamma w), w \in [0, \beta_1)$ and $\phi_P(w) = U, w \in [\beta_1, 1]$ (i.e., green curve) since $\beta_2 = \beta_1' = 1$ and $M_1 = U$ by solving equation (10) with $P = U$. This threshold is $(\eta, \gamma)$-competitive over $[U]$ and $[L, U)$. When the prediction is in the intermediate region $P \in [M, U)$, the threshold function consists of at most four segments. The first and the forth segments when $w \in [0, \beta_1)$ and $w \in (\beta_2, 1]$ are exponential functions with rate $\gamma$, aiming to ensure $\gamma$-competitiveness over $[L, M_1)$ and $(P, U]$. These two segments may be absorbed when the prediction is small ($M_1 \leq L\gamma$) or large ($P\gamma/\eta \geq U$), corresponding to $\beta_1 = 0$ and $\beta_2 = 1$. To guarantee a good consistency, a flat segment in $w \in [\beta_1, \beta_1')$ is designed to convert enough dollar before reaching the price $P$ by enforcing $P \geq M_1$, and an exponential segment with rate $\eta$ in $w \in [\beta_1', \beta_2]$ to ensure $\eta$-competitiveness over $[M_1, P]$. A full proof can be found in Appendix A.5.

## 5 Pareto-optimal consistency-robustness trade-off

To this point, we have focused on upper bounds for robustness and consistency. This section provides lower bounds on the robustness-consistency trade-offs for both 1-max-search and one-way trading and shows the Pareto-optimality of our proposed learning-augmented algorithms. Note that, obtaining lower bounds on the trade-off between robustness and consistency for online algorithms has proven difficult. The only existing tight lower bounds we are aware of are in the case of deterministic [1] and randomized [4, 22] algorithms for the ski-rental problem.

**Theorem 5.1.** *Any $\gamma$-robust deterministic* LOA *for 1-max-search must have consistency $\eta \geq \theta/\gamma$. Thus,* OTA *with the reservation price* (6) *is Pareto-optimal.*

**Theorem 5.2.** *If a deterministic* LOA *for one-way trading is $\gamma$-robust, its consistency is at least $\eta \geq \theta/[\frac{\theta}{\gamma} + (\theta - 1)(1 - \frac{1}{\gamma}\ln\frac{\theta - 1}{\gamma - 1})]$. Thus,* OTA *with the threshold function* (8) *is Pareto-optimal.*

We illustrate the Pareto-optimal trade-offs of robustness and consistency for 1-max-search and one-way trading in Figure 2. Notice that the Pareto-boundary of one-way trading dominates that of 1-max-search since the fractional conversion leaves more flexibility to online decisions in one-way trading, leading to a better lower bound. For both problems, with the improvement of consistency from the optimal competitive ratio (i.e., $\sqrt{\theta}$ or $\alpha^*$) to the best possible ratio 1, the robustness degrades from the optimal competitive ratio to the worst possible ratio $\theta$. This means there is no free lunch in online conversion problems; to achieve a good consistency, robustness must be sacrificed.

We end the section by proving Theorem 5.2 for one-way trading. A proof of Theorem 5.1 is included in Appendix A.6.

**Proof of Theorem 5.2.** To show a lower bound result, we first construct a special family of instances, and then show that for any $\gamma$-robust LOA (not necessarily being OTA), their consistency $\eta$ is lower bounded under the special instances.

We focus on a collection of $p$-instances $\{\mathcal{I}_p\}_{p \in [L,U]}$ where $p$ ranges from $L$ to $U$, where a $p$-instance is defined as follows.

**Definition 5.3** ($p$-instance). *Given $p \in [L, U]$ and a large $N$, an instance $\mathcal{I}_p := \{v_1, \ldots, v_N\}$ is called a $p$-instance if $v_n = L + (n - 1)\delta, n \in [N - 1]$ with $\delta = \frac{p - L}{N - 2}$ and $v_N = L$.*

Notice that, when $N \to \infty$, the sequence of prices in $\mathcal{I}_p$ continuously increases from $L$ to $p$, and drops to $L$ in the last step.

Let $g(p) : [L, U] \to [0, 1]$ denote a conversion function of a deterministic LOA for one-way trading, where $g(p)$ is its total amount of converted dollar under the instance $\mathcal{I}_p$ before the compulsory conversion in the last step. A key observation is that for a large $N$, executing the instance $\mathcal{I}_{p+\delta}$ is equivalent to first executing $\mathcal{I}_p$ (excluding the last step) and then processing $p + \delta$ and $L$. Since the conversion decision is unidirectional and deterministic, we must have $g(p + \delta) \geq g(p)$, i.e., $g(p)$ is non-decreasing in $[L, U]$. In addition, the whole dollar must be converted once the maximum price $U$ is observed, i.e., $g(U) = 1$.

Under the instance $\mathcal{I}_p$, the offline optimal profit is $\text{OPT}(\mathcal{I}_p) = p$ and the profit of an online algorithm with conversion function $g$ is $\text{ALG}(\mathcal{I}_p) = g(L)L + \int_L^p udg(u) + L(1 - g(p))$, where $udg(u)$ is the profit of converting $dg(u)$ dollar at the price $u$. The first two terms are the cumulative profit before the last step and the last term is from the compulsory conversion.

For any $\gamma$-robust online algorithm, the corresponding conversion function must satisfy $\text{ALG}(\mathcal{I}_p) \geq \text{OPT}(\mathcal{I}_p)/\gamma = p/\gamma, \forall p \in [L, U]$. If, additionally, given prediction $P \geq \gamma L$, no dollar needs to be converted under instances $\{\mathcal{I}_p\}_{p \in [L, \gamma L)}$, i.e., $g(p) = 0, \forall p \in [L, \gamma L)$. This is because if a $\gamma$-robust online algorithm converts any dollar below the price $\gamma L$, we can always design a new algorithm by letting it convert the dollar at the price $\gamma L$ instead. The new online algorithm is still $\gamma$-robust and achieves a smaller consistency when the prediction is accurate. Thus, given $P = U \geq L\gamma$, the conversion function of any $\gamma$-robust online algorithm must satisfy

$$\text{ALG}(\mathcal{I}_p) = g(\gamma L)\gamma L + \int_{\gamma L}^p udg(u) + L(1 - g(p)) \geq \frac{p}{\gamma}, \quad \forall p \in [\gamma L, U]. \quad (11)$$

By integral by parts, a necessary condition for above robustness constraint (11) to hold is $g(p) \geq \frac{p/\gamma - L}{p - L} + \frac{1}{p - L} \int_{\gamma L}^p g(u)du$. Based on Gronwall's Inequality (see Theorem 1, p.356, [17]), we have

$$g(p) \geq \frac{p/\gamma - L}{p - L} + \frac{1}{\gamma} \int_{\gamma L}^p \frac{u - \gamma L}{(u - L)^2} du = \frac{1}{\gamma} \ln \frac{p - L}{\gamma L - L}, \quad \forall p \in [\gamma L, U]. \quad (12)$$

In addition, to ensure $\eta$-consistency when the prediction is $P = U$, we must ensure $\text{ALG}(\mathcal{I}_U) \geq \text{OPT}(\mathcal{I}_U)/\eta$. Combining this constraint with $g(U) = 1$ gives

$$\int_{\gamma L}^U g(u)du \leq (\eta - 1)U/\eta. \quad (13)$$

By combining equations (12) and (13), the conversion function $g(p)$ of any $\gamma$-robust and $\eta$-consistent online algorithm given $P = U$ must satisfy $\int_{\gamma L}^U \frac{1}{\gamma} \ln \frac{u - L}{\gamma L - L} du \leq \int_{\gamma L}^U g(u)du \leq (\eta - 1)U/\eta$, which equivalently gives $\eta \geq \theta / [\frac{\theta}{\gamma} + (\theta - 1)(1 - \frac{1}{\gamma} \ln \frac{\theta - 1}{\gamma - 1})]$.

Finally, since Theorem 4.6 has shown that OTA with the threshold function (8) can achieve the lower bound in Theorem 5.2, it is Pareto-optimal.

# 6 Numerical results

We end with a case study on the exchange of Bitcoin (BTC) to USD. This case study is timely since the rapid growth of cryptocurrency has left many traders eager to profit from rising and falling of exchange rates in average-case scenarios, while the uncertainty and volatility of cryptocurrency have made many traders cautious of unforeseeable crashes in worst-case scenarios. Our results answer two questions: (Q1) *How does the learning-augmented* OTA *compare to pure online algorithms with different prediction qualities and drastic exchange rate crashes?* (Q2) *How should the* OTA *select the robustness parameter λ, and especially, would an online learning algorithm work in practice?*

**Setup.** We use historical BTC prices in USD of 5 years from 2015 to 2020, with exchange rates collected every 5 minutes from the Gemini exchange. We assume one BTC is available for trading during 250 instances of length one week. Since BTC is traded in the unit of satoshi (i.e., 0.00000001 BTC), this problem fits the fractional conversion setting and we apply the LOA for the one-way trading problem in the following experiments. We set $L$ and $U$ as the historical minimum and maximum prices over the entire 5 years.[1] To generate a prediction $P$, we simply use the observed maximum

---

[1]The focus of this paper is not on the impact of $L$ and $U$, and thus we simply set them as historical values. In practical trading problems, $L$ and $U$ can be considered as predetermined parameters that represent the stop-loss and take-profit prices in the exit strategy of the trading process.

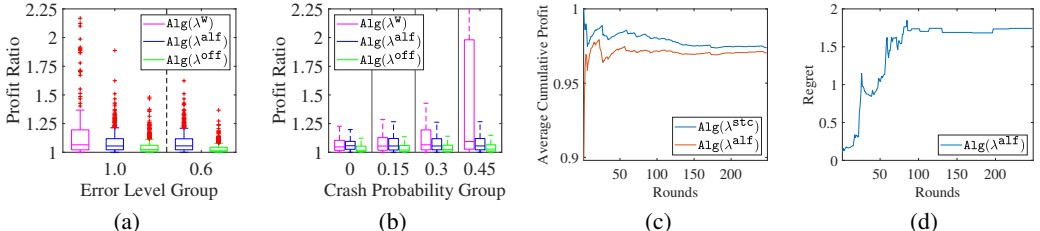

Figure 3: Profit ratios of different algorithms with (a) different prediction errors and (b) different crash probabilities. The evolution of (c) the average cumulative profit and (d) regret of $\texttt{Alg}(\lambda^{\texttt{alf}})$

exchange rate of the previous week. To evaluate the impact of prediction quality, we adjust the error level between 0.0 to 1.0, where 0.0 indicates perfect predictions and 1.0 indicates unadjusted predictions. To evaluate the performance in worst-case settings, we also introduce a *crash probability* $q$, where the exchange rate of BTC at the last slot will crash to $L$ with probability $q$.

We compare the empirical profit ratio of four different algorithms: (i) $\texttt{Alg}(\lambda^{\texttt{w}})$, the worst-case optimized online algorithm that does not take into account predictions, but, guarantees the optimal competitive ratio; (ii) $\texttt{Alg}(\lambda^{\texttt{off}})$, an algorithm that finds the best possible distrust parameter $\lambda$ in an offline manner; this algorithm is not practical since it is fed with the optimal parameter; however, it illustrates the largest possible improvement from predictions under our algorithm; (iii) $\texttt{Alg}(\lambda^{\texttt{alf}})$, an online learning algorithm from [16] which selects the parameter using the adversarial Lipschitz algorithm in a full-information setting; and (iv) $\texttt{Alg}(\lambda^{\texttt{stc}})$, an online algorithm that uses the best static $\lambda$ and serves as the baseline for $\texttt{Alg}(\lambda^{\texttt{alf}})$. Additional details are in the supplementary material.

**Experimental results.** We answer Q1 in Figures 3(a) and 3(b), and Q2 in Figures 3(c) and 3(d). Figure 3(a) compare the profit ratios of several algorithms at different error levels. First, it shows that $\texttt{Alg}(\lambda^{\texttt{off}})$ and $\texttt{Alg}(\lambda^{\texttt{alf}})$ noticeably improve the performance of $\texttt{Alg}(\lambda^{\texttt{w}})$. The upper boxplot whisker of $\texttt{Alg}(\lambda^{\texttt{w}})$ is 1.35, while $\texttt{Alg}(\lambda^{\texttt{alf}})$ at 1.0 error level has an upper boxplot whisker around 1.25. Second, it shows that the gap between $\texttt{Alg}(\lambda^{\texttt{off}})$ and $\texttt{Alg}(\lambda^{\texttt{alf}})$ is quite small, as the upper boxplot whisker of $\texttt{Alg}(\lambda^{\texttt{off}})$ is slightly lower. Comparing the profit ratios of different algorithms with different crash probability values at 1.0 error level in Figure 3(b), we see that the performance of $\texttt{Alg}(\lambda^{\texttt{w}})$ drastically degrades at crash probability 0.45. However, both $\texttt{Alg}(\lambda^{\texttt{alf}})$ and $\texttt{Alg}(\lambda^{\texttt{off}})$ are stable at high crash probability. Figure 3(c) compares the average normalized profit of $\texttt{Alg}(\lambda^{\texttt{alf}})$ and $\texttt{Alg}(\lambda^{\texttt{stc}})$ and shows the reward of $\texttt{Alg}(\lambda^{\texttt{alf}})$ converges toward that of $\texttt{Alg}(\lambda^{\texttt{stc}})$ as the learning process moves forward. Figure 3(d) indicates that the regret of $\texttt{Alg}(\lambda^{\texttt{alf}})$ stabilizes.

## 7 Concluding remarks

To improve upon the performance of algorithms for online conversion problems that are designed with worst-case guarantees in mind, this paper has incorporated machined-learned predictions into the design of a class of $\texttt{OTA}$ and shown that the learning-augmented $\texttt{OTA}$ can achieve Pareto-optimal robustness-consistency trade-offs. This result represents only the second tight lower bound result in the robustness and consistency analysis of $\texttt{LOA}$, with the first being for ski rental problem [1, 4, 22]. We expect that our method of deriving lower bounds can be extended to more general online optimization problems with capacity constraints. A limitation of this work is that consistency and robustness only measure the competitive performance in two extreme cases, i.e., the predictions are perfectly accurate or completely wrong. Although our approach for design and analysis of $\texttt{OTA}$ provides opportunities to design $\texttt{OTA}$ in ways that guarantee more fine-grained performance metrics, this is left for the future work. Another limitation is that this work provides only empirical evaluation of the algorithm $\texttt{Alg}(\lambda^{\texttt{alf}})$ that selects the robustness parameter in an online manner. Deriving theoretical bounds remains open. Last, we cannot see any negative societal impacts of our work.

## Acknowledgements

Bo Sun and Danny H.K. Tsang acknowledge the support received from the Hong Kong Research Grant Council (RGC) General Research Fund Project 16211220. Russell Lee and Mohammad Hajiesmaili's research is supported by NSF CAREER 2045641, CNS 2106299, CNS 2102963, CNS 1908298. Adam Wierman's research is funded by NSF CNS-2106403 and NGSDI-2105648. We also acknowledge the support received from the Resnick Institute and the Center for Autonomous Systems and Technologies at Caltech.

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
