# Appendix A   Technical Proofs

## A.1   Proof of Proposition 4.1

To see the robustness of OTA with the reservation price $\Phi_P = \lambda\sqrt{LU} + (1-\lambda)P$ for 1-max-search, we consider the following two cases.

**Case I.** When the actual maximum price $V$ is smaller than the reservation price, i.e., $V < \Phi_P$, 1 dollar is converted at the last step with the worst possible price $L$, and thus the worst-case ratio is

$$\frac{\mathtt{OPT}(\mathcal{I})}{\mathtt{ALG}(\mathcal{I})} = \frac{V}{L} < \frac{\Phi_P}{L} \leq \lambda\sqrt{\theta} + (1-\lambda)\theta, \tag{14}$$

where the last inequality is due to $P \leq U$.

**Case II.** When $V \geq \Phi_P$, the worst-case ratio is

$$\frac{\mathtt{OPT}(\mathcal{I})}{\mathtt{ALG}(\mathcal{I})} = \frac{V}{\Phi_P} \leq \frac{\theta}{\lambda\sqrt{\theta} + 1 - \lambda}, \tag{15}$$

where the inequality is due to $V \leq U$ and $P \geq L$.

Thus, the robustness is $\max\{\lambda\sqrt{\theta} + (1-\lambda)\theta, \theta/(\lambda\sqrt{\theta} + 1 - \lambda)\} = \lambda\sqrt{\theta} + (1-\lambda)\theta$.

To see the consistency, consider the following two cases when the prediction is accurate, i.e., $P = V$.

**Case I.** When $P \geq \sqrt{LU}$, we have $V = P \geq \Phi_P$ for $\lambda \in [0, 1]$, and the worst-case ratio is

$$\frac{\mathtt{OPT}(\mathcal{I})}{\mathtt{ALG}(\mathcal{I})} = \frac{V}{\Phi_P} = \frac{V}{\lambda\sqrt{LU} + (1-\lambda)V} \leq \frac{\theta}{\lambda\sqrt{\theta} + (1-\lambda)\theta}. \tag{16}$$

**Case II.** When $P < \sqrt{LU}$, we have $V = P < \Phi_P$ for $\lambda \in (0, 1]$, and the worst-case ratio

$$\frac{\mathtt{OPT}(\mathcal{I})}{\mathtt{ALG}(\mathcal{I})} = \frac{V}{L} < \frac{\lambda\sqrt{LU} + (1-\lambda)V}{L} < \sqrt{\theta}. \tag{17}$$

Combining above two cases gives the consistency $\max\{\sqrt{\theta}, \theta/(\lambda\sqrt{\theta} + (1-\lambda)\theta)\} = \sqrt{\theta}$ for $\lambda \in (0, 1]$. In the special case when $\lambda = 0$, $\Phi_P = P = V$ and the consistency is 1.

## A.2   Proof of Lemma 4.3

Recall that given a prediction $P$ on the maximum price $V$ over the price sequence $\mathcal{I}$, the prediction error is $\varepsilon = 0$ if $\mathcal{I} \in \Omega_P$. Thus, we have $\mathtt{CR}(0) = \max_{\mathcal{I} \in \Omega_P} \mathtt{OPT}(\mathcal{I})/\mathtt{ALG}(\mathcal{I})$. In addition, arbitrary prediction errors mean that the actual instance $\mathcal{I}$ can take any possible instances in $\Omega$. Therefore, we have $\max_\varepsilon \mathtt{CR}(\varepsilon) = \max_{\mathcal{I} \in \Omega} \mathtt{OPT}(\mathcal{I})/\mathtt{ALG}(\mathcal{I})$.

If OTA is $(\eta, \gamma)$-competitive over a partition $\{\mathcal{P}_\eta, \mathcal{P}_\gamma\}$ with $\Omega = \mathcal{P}_\eta \cup \mathcal{P}_\gamma$ and $\Omega_P \subseteq \mathcal{P}_\eta$, based on the definition of the generalized competitive ratio, we have

$$\gamma \geq \max\{\eta, \gamma\} = \max_{\mathcal{I} \in \mathcal{P}_\eta \cup \mathcal{P}_\gamma} \frac{\mathtt{OPT}(\mathcal{I})}{\mathtt{ALG}(\mathcal{I})} = \max_\varepsilon \mathtt{CR}(\varepsilon), \tag{18}$$

$$\eta = \max_{\mathcal{I} \in \mathcal{P}_\eta} \frac{\mathtt{OPT}(\mathcal{I})}{\mathtt{ALG}(\mathcal{I})} \geq \max_{\mathcal{I} \in \Omega_P} \frac{\mathtt{OPT}(\mathcal{I})}{\mathtt{ALG}(\mathcal{I})} = \mathtt{CR}(0). \tag{19}$$

Thus, OTA is $\eta$-consistent and $\gamma$-robust.

## A.3   Proof of Theorem 4.4

Our goal is to prove that OTA with the threshold function $\phi := \{\phi_i\}_{i \in [I]}$ is $\alpha_i$-competitive over subset $\mathcal{P}_i = \{\Omega_p\}_{p \in [M_{i-1}, M_i)}$ for all $i \in [I]$ when each piece of the threshold function, $\phi_i$, satisfies the condition in one of the cases of Theorem 4.4. Consider the instance subset in the following three cases, corresponding to the three cases of the sufficient condition in Theorem 4.4.

**Case I.** $\mathcal{P}_i$ with $M_i \leq \phi(0)$. For any instance $\mathcal{I} \in \mathcal{P}_i$, the maximum price is smaller than $M_i$ and hence $\mathtt{OPT}(\mathcal{I}) < M_i$. Since $\beta_i = 0 = \beta_{i-1}$, this threshold piece is absorbed. Because $\phi$ is a right-continuous function, the reservation price at $\beta_i$ is no smaller than $M_i$. Thus, $\mathtt{OTA}$ converts no dollar when executing $\mathcal{I}$, excluding the compulsory conversion in the last step, and its return is $\mathtt{ALG}(\mathcal{I}) = v_N \geq L$. In this case, when the threshold function $\phi_i$ and the competitive ratio $\alpha_i$ satisfy $M_i \leq \alpha_i L$, $\mathtt{OPT}(\mathcal{I})/\mathtt{ALG}(\mathcal{I}) < M_i/L \leq \alpha_i, \forall \mathcal{I} \in \mathcal{P}_i$.

**Case II.** $\mathcal{P}_i$ with $\phi(0) < M_i \leq \phi(1)$. Without loss of generality, we only consider the instances whose price in the last step is not the unique maximum price. This is because we can instead consider an alternative instance, which appends an additional maximum price just after the unique maximum price as the last price, and this alternative instance leads to the same offline and online returns as the original instance when being executed by $\mathtt{OTA}$. Thus, we can use $w^{(N-1)}$ to denote the final utilization of $\mathtt{OTA}$ after executing any instance.

Under an instance $\mathcal{I} \in \Omega_p \subseteq \mathcal{P}_i$ with a maximum price $p$, the return of offline optimal is $\mathtt{OPT}(\mathcal{I}) = p$, and the return of $\mathtt{OTA}$ is

$$\mathtt{ALG}(\mathcal{I}) = \sum_{n \in [N-1]} v_n \bar{x}_n + (1 - w^{(N-1)})v_N \tag{20}$$

$$\geq \sum_{n \in [N-1]} \int_{w^{(n-1)}}^{w^{(n)}} \phi(u)du + (1 - w^{(N-1)})L \tag{21}$$

$$= \int_0^{w^{(N-1)}} \phi(u)du + (1 - w^{(N-1)})L, \tag{22}$$

where $\mathtt{ALG}(\mathcal{I})$ consists of the return of conversions $\{\bar{x}_n\}_{n \in [N-1]}$ by $\mathtt{OTA}$ and the compulsory conversion $1 - w^{(N-1)}$ in the last step. The inequality (21) holds since (i) $v_N \geq L$, and (ii) $\bar{x}_n$ is the optimal solution of the optimization Line 3 in Algorithm 1, which ensures $v_n \bar{x}_n \geq \int_{w^{(n-1)}}^{w^{(n)}} \phi(u)du$.

In this case, $\phi_i$ is in the form of Equation (2), which consists of a flat segment in $[\beta_{i-1}, \beta'_{i-1})$ and an increasing segment $\varphi_i(w)$ in $[\beta'_{i-1}, \beta_i)$ that satisfies the differential equation (3). Note that $w^{(N-1)}$ is the final utilization of $\mathtt{OTA}$ after executing $\mathcal{I} \in \Omega_p$ and $w^{(N-1)} \in [\beta'_{i-1}, \beta_i)$. Also noticing that $\phi$ can be discontinuous at $w = \beta'_{i-1}$, we further consider two sub-cases.

**Case II(a).** if $M_{i-1} \leq p < \varphi_i(\beta'_{i-1})$, then $w^{(N-1)} = \beta'_{i-1}$. In this case, we have

$$\frac{\mathtt{OPT}(\mathcal{I})}{\mathtt{ALG}(\mathcal{I})} \leq \frac{p}{\int_0^{w^{(N-1)}} \phi(u)du + (1 - w^{(N-1)})L} < \frac{\varphi_i(\beta'_{i-1})}{\int_0^{\beta'_{i-1}} \phi(u)du + (1 - \beta'_{i-1})L} \leq \alpha_i,$$

where the last inequality is due to the differential equation (3) at $w = \beta'_{i-1}$.

**Case II(b).** if $\varphi_i(\beta'_{i-1}) \leq p < \varphi_i(\beta_i) = M_i$, we have $p = \varphi_i(w^{(N-1)})$ and

$$\frac{\mathtt{OPT}(\mathcal{I})}{\mathtt{ALG}(\mathcal{I})} \leq \frac{p}{\int_0^{w^{(N-1)}} \phi(u)du + (1 - w^{(N-1)})L} = \frac{\varphi_i(w^{(N-1)})}{\int_0^{w^{(N-1)}} \phi(u)du + (1 - w^{(N-1)})L} \leq \alpha_i,$$

where the last inequality is also due to the differential equation (3).

Combining the above two sub-cases gives $\mathtt{OPT}(\mathcal{I})/\mathtt{ALG}(\mathcal{I}) \leq \alpha_i, \forall \mathcal{I} \in \Omega_p, \forall \Omega_p \subseteq \mathcal{P}_i$.

**Case III.** $\mathcal{P}_i$ with $M_i > \phi(1)$. Since $\phi(1)$ is one of the price segment boundaries, $M_{i-1} \geq \phi(1)$ and hence $\beta_{i-1} = 1$. Thus, $\phi_i$ is absorbed. For any instance $\mathcal{I} \in \mathcal{P}_i$, its maximum price is no smaller than the maximum value of the threshold function $\phi(1)$, and thus the whole dollar will be converted to yens before the compulsory conversion, i.e., $w^{(N-1)} = 1$. The return of the offline optimal is $\mathtt{OPT}(\mathcal{I}) \leq M_i$ and the return of $\mathtt{OTA}$ is $\mathtt{ALG}(\mathcal{I}) \geq \int_0^1 \phi(u)du$. In this case, when $\phi$ and $\alpha_i$ satisfy $M_i \leq \alpha_i \int_0^1 \phi(u)du$, we have $\mathtt{OPT}(\mathcal{I})/\mathtt{ALG}(\mathcal{I}) \leq M_i/\int_0^1 \phi(u)du \leq \alpha_i, \forall \mathcal{I} \in \mathcal{P}_i$.

In summary, $\mathtt{OPT}(\mathcal{I})/\mathtt{ALG}(\mathcal{I}) \leq \alpha_i, \forall \mathcal{I} \in \mathcal{P}_i$ and thus $\mathtt{OTA}$ is $\alpha_i$-competitive over $\mathcal{P}_i, \forall i \in [I]$ if each piece of the threshold function of $\mathtt{OTA}$ satisfies one of the sufficient conditions in Theorem 4.4.

## A.4 Proof of Theorem 4.5

We show the competitiveness of `OTA` with the reservation price (6) for 1-max-search based on the sufficient condition in Theorem 4.4 and further prove the robustness and consistency bounds based on Lemma 4.3. Consider the following three cases.

**Case I.** Given $P \in [L, L\eta)$, we consider a partition $\mathcal{P} := \{[L, L\eta), [L\eta, U]\}$ by letting $M_1 = L\eta$ and $M_2 = U$, and aim to show that `OTA` is $(\eta, \gamma)$-competitive over $[L, L\eta)$ and $[L\eta, U]$. In this case, we have $\Phi_P = L\eta = \phi(0) = \phi(1)$. Therefore, the subset $[L, L\eta)$ belongs to Case I of the sufficient condition. Since $M_1/L \leq \eta$, `OTA` is $\eta$-competitive over $[L, L\eta)$. For subset $[L\eta, U]$, we have $M_2 = U > \phi(1)$. Based on the sufficient condition in Case III of Theorem 4.4 and knowing that $M_2 / \int_0^1 \Phi_P du = U/(L\eta) = \theta/\eta = \gamma$, `OTA` is $\gamma$-competitive over $[L\eta, U]$. Thus, `OTA` is $(\eta, \gamma)$-competitive over $\mathcal{P}$.

**Case II.** Given $P \in [L\eta, L\gamma)$, we consider a partition $\mathcal{P} := \{[L, \Phi_P), [\Phi_P, P], (P, U]\}$ by letting $M_1 = \Phi_P$, $M_2 = P$ and $M_3 = U$. Note that $P \geq \Phi_P$ since $P = \eta P/\eta = \lambda\gamma P/\eta + (1-\lambda)P/\eta \geq \Phi_P$ based on the design of $\eta$ and $\gamma$ in Equation (5). We aim to show that `OTA` is $(\gamma, \eta, \gamma)$-competitive over $\mathcal{P}$. In this case, $\phi(0) = \phi(1) = \Phi_P = \lambda L\gamma + (1-\lambda)P/\eta$.

For subset $[L, \Phi_P)$, we have $M_1 \leq \phi(0)$ and we consider the sufficient condition in Case I. Since

$$\frac{M_1}{L} = \frac{\Phi_P}{L} = \lambda\gamma + \frac{1-\lambda}{L\eta}P \leq \lambda\gamma + (1-\lambda)\frac{\gamma}{\eta} \leq \gamma, \tag{23}$$

`OTA` is $\gamma$-competitive over $[L, \Phi_P)$.

For subset $[\Phi_P, P]$, if $P \in (L\eta, L\gamma)$, we have $M_2 = P > \phi(1) = \Phi_P$ and consider the sufficient condition in Case III. Since

$$\frac{M_2}{\int_0^1 \Phi_P du} = \frac{P}{\Phi_P} = \frac{P}{\lambda L\gamma + \frac{1-\lambda}{\eta}P} \leq \frac{1}{\lambda + \frac{1-\lambda}{\eta}} \leq \eta, \tag{24}$$

`OTA` is $\eta$-competitive over $[\Phi_P, P]$. If $P = L\eta$, we have $M_2 = \phi(0)$. Based on Case I of the sufficient condition and $M_2/L = \eta$, `OTA` is also $\eta$-competitive over $[\Phi_P, P]$.

For subset $(P, U]$, we consider the sufficient condition of Case III, and we have

$$\frac{M_3}{\int_0^1 \Phi_P du} = \frac{U}{\Phi_P} = \frac{U}{\lambda L\gamma + \frac{1-\lambda}{\eta}P} \leq \frac{U}{\lambda L\gamma + (1-\lambda)L} = \frac{\theta}{\eta} = \gamma, \tag{25}$$

where we apply $\eta = \lambda\gamma + 1 - \lambda$ in (5). Therefore, `OTA` is $\gamma$-competitive over $(P, U]$.

Thus, `OTA` is $(\gamma, \eta, \gamma)$-competitive over $\mathcal{P}$.

**Case III.** Given $P \in [L\gamma, U]$, we consider a partition $\mathcal{P} := \{[L, L\gamma), [L\gamma, U]\}$ by letting $M_1 = L\gamma$, and aim to show `OTA` is $(\gamma, \eta)$-competitive over $\mathcal{P}$. For subsets $[L, L\gamma)$ and $[L\gamma, U]$, we have $M_1/L = \gamma$ and $U/\Phi_P = U/(L\gamma) = \eta$, respectively. Based on the sufficient condition in Case I and Case III, `OTA` is $(\gamma, \eta)$-competitive over $\mathcal{P}$.

In above three cases, given any $P$, we have shown that there exists a partition and `OTA` is $\eta$-competitive for the instance subset that contains $\Omega_P$ and $\gamma$-competitive for the other subsets. Based on Lemma 4.3, `OTA` is $\eta$-consistent and $\gamma$-robust.

## A.5 Proof of Theorem 4.6

We prove the competitiveness of `OTA` with the threshold function in (8) for one-way trading based on the sufficient condition in Theorem 4.4. Consider the following three cases.

**Case I.** Given $P \in [L, M)$, we consider a partition $\mathcal{P} := \{[L, M), [M, U]\}$, and aim to show that `OTA` is $(\eta, \gamma)$-competitive over $[L, M)$ and $[M, U]$. Let $\phi_1(w), w \in [0, \beta)$ and $\phi_2(w), w \in [\beta, 1]$ denote the two pieces of the threshold functions given in (8a). Both $[L, M)$ and $[M, U]$ belong to Case II of the sufficient condition in Theorem 4.4.

For subset $[L, M)$, $\phi_1(w) = L + (L\eta - L)\exp(\eta w)$ has no flat segment, i.e., $\beta_0' = \beta_0 = 0$, and its increasing segment $\varphi_1$ is the solution of the differential equation

$$\begin{cases} \varphi_1(w) = \eta \left[ \int_0^w \varphi_1(u)du + (1-w)L \right], w \in [0, \beta), \\ \varphi_1(\beta) = M, \end{cases} \tag{26}$$

which satisfies the sufficient condition in (3) if $M = L + (L\eta - L)\exp(\eta\beta)$.

For subset $[M, U]$, $\phi_2(w) = L + (U - L)\exp(\gamma(w - 1))$ also has no flat segment, i.e., $\beta_1 = \beta_1' = \beta$, and the increasing segment $\varphi_2$ is the solution of

$$\begin{cases} \varphi_2(w) = \gamma\left[\int_0^\beta \phi(u)du + \int_\beta^w \varphi_2(u)du + (1-w)L\right], w \in [\beta, 1], \\ \varphi_2(1) = U, \end{cases} \tag{27}$$

which satisfies the sufficient condition in (3) if $M\gamma/\eta = L + (U - L)\exp(\gamma(\beta - 1))$.

Since $M$ and $\beta$ are the solution of equation (9), both $\phi_1$ and $\phi_2$ satisfy the sufficient condition in Case II. Thus, OTA is $(\eta, \gamma)$-competitive over $[L, M)$ and $[M, U]$

**Case II.** Given $P \in [M, U)$, we consider a partition $\mathcal{P} := \{[L, M_1], [M_1, P], (P, U]\}$, where $M_2 = P$, $M_3 = U$, and $M_1 \in [M, P)$ is to be determined. We aim to show that OTA with $\phi$ in (8b) is $(\gamma, \eta, \gamma)$-competitive over $\mathcal{P}$. Let $\phi_1(w), w \in [0, \beta_1)$, $\phi_2(w), w \in [\beta_1, \beta_2]$, and $\phi_3(w), w \in (\beta_2, 1]$ denote the threshold pieces corresponding to the three subsets. Based on the threshold function (8b) and $\beta_1$ and $\beta_2$ determined in equation (10), we have $\phi(0) = \min\{M_1, L\gamma\}$, and $\phi(1) = P$ if $P\gamma/\eta \geq U$ and $\phi(1) = U$ if $P\gamma/\eta < U$.

For subset $[L, M_1)$, we consider the following two sub-cases based on the value of $\max\{M_1/L, \gamma\}$.

**Case II(a).** If $M_1 \leq L\gamma$, based on the first equation in (10), $\beta_1 = 0$ and hence $\phi_1$ is absorbed. In this case, $M_1 = \phi(0)$ and thus $[L, M_1)$ belongs to Case I of the sufficient condition. Since $M_1/L \leq \gamma$, OTA is $\gamma$-competitive over $[L, M_1)$.

**Case II(b).** If $M_1 > L\gamma$, we have $\phi(0) < M_1 \leq \phi(1)$ and hence $[L, M_1)$ belongs to Case II of the sufficient condition. In this case, $\phi_1(w) = L + (\gamma L - L)\exp(\gamma w), w \in [0, \beta_1)$ has no flat segment, and the increasing segment $\varphi_1$ is the solution of

$$\begin{cases} \varphi_1(w) = \gamma\left[\int_0^w \varphi_1(u)du + (1-w)L\right], w \in [0, \beta_1), \\ \varphi_1(\beta_1) = M_1, \end{cases} \tag{28}$$

which satisfies the sufficient condition in (3) if $M_1 = L + (\gamma L - L)\exp(\gamma\beta_1)$.

Summarizing Case II(a) and Case II(b), OTA is $\gamma$-competitive over $[L, M_1)$ if the first equation in (10) holds.

For subset $[M_1, P]$, since $\phi(0) < P \leq \phi(1)$, $[M_1, P]$ belongs to Case II of the sufficient condition. $\phi_2(w) = M_1, w \in [\beta_1, \beta_1')$ is a flat segment and the sufficient condition in (3) holds when $w = \beta_1'$ if the length of this segment ensures $\phi_2(\beta_1') = M_1 = \eta[\int_0^{\beta_1} \phi(u)du + (\beta_1' - \beta_1)M_1 + (1 - \beta_1')L]$, which is the second equation in (10).

The increasing segment $\varphi_2(w), w \in [\beta_1', \beta_2]$ is the solution of

$$\begin{cases} \varphi_2(w) = \eta\left[\int_0^{\beta_1} \phi(u)du + (\beta_1' - \beta_1)M_1 + \int_{\beta_1'}^w \varphi_2(u)du + (1-w)L\right], w \in [\beta_1', \beta_2], \\ \varphi_2(\beta_2) = P, \end{cases} \tag{29}$$

which satisfies the sufficient condition in (3) if $P = L + (M_1 - L)\exp(\eta(\beta_2 - \beta_1'))$. Thus, OTA is $\eta$-competitive over $[M_1, P]$ if the second and third equations in (10) hold.

For subset $(P, U]$, we have the following two sub-cases based on the value of $\min\{P\gamma/\eta, U\}$.

**Case II(c).** If $P\gamma/\eta < U$, we have $\phi(0) < U \leq \phi(1) = U$ and thus $(P, U]$ belongs to Case II of the sufficient condition. $\phi_3(w) = L + (U - L)\exp(\gamma(w - 1)), w \in (\beta_2, 1]$ has no flat segment and its increasing segment $\varphi_3$ is the solution of

$$\begin{cases} \varphi_3(w) = \gamma\left[\int_0^{\beta_2} \phi(u)du + \int_{\beta_2}^w \varphi_3(u)du + (1-w)L\right], w \in (\beta_2, 1], \\ \varphi_3(1) = U, \end{cases} \tag{30}$$

which satisfies the sufficient condition (3) if $P\gamma/\eta = L + (U - L)\exp(\gamma(\beta_2 - 1))$.

**Case II(d).** If $P\gamma/\eta \geq U$, we have $\beta_2 = 1$ based on the fourth equation in (10). Thus, $\phi(1) = P < U$ and $(P, U]$ belongs to Case III of the sufficient condition. Since $U/\int_0^1 \phi(u)du = U\eta/P \leq \gamma$, OTA is $\gamma$-competitive over $(P, U]$.

Based on Case II(c) and Case II(d), OTA is $\gamma$-competitive over $(P, U]$ if the forth equation in (10) holds.

In summary, since $M_1$, $\beta_1$, $\beta_1'$ and $\beta_2$ are the solution of equation (10), OTA with the threshold function (8b) is $(\gamma, \eta, \gamma)$-competitive over $[L, M_1), [M_1, P], (P, U]$.

**Case III.** Given $P = U$, we consider a partition $\mathcal{P} := \{[L, U), [U]\}$. Based on equation (10) with $P = U$, we have $\beta_2 = \beta_1' = 1$ and $M_1 = U$. Thus, $\phi_3$ and the increasing segment of $\phi_2$ are absorbed. Since $\phi(0) < U \leq \phi(1)$, both $[L, U)$ and $[U]$ belong to Case II of the sufficient condition.

For subset $[L, U)$, $\phi_1(w) = L + (\gamma L - L) \exp(\gamma w)$, $w \in [0, \beta_1)$ has no flat segment and its increasing segment $\varphi_1$ is the solution of

$$\begin{cases} \varphi_1(w) = \gamma \left[ \int_0^w \varphi_1(u) du + (1 - w)L \right], w \in [0, \beta_1), \\ \varphi_1(\beta_1) = U, \end{cases} \tag{31}$$

which satisfies the sufficient condition (3) if $M_1 = U = L + (\gamma L - L) \exp(\gamma \beta_1)$.

For subset $[U]$, $\phi_2(w) = U, w \in [\beta_1, 1]$ satisfies the sufficient condition in (3) when $w = \beta_1' = 1$ if $\varphi_2(1) = U = \eta[\int_0^{\beta_1} \phi(u) du + (1 - \beta_1)U]$. This equation holds if the first equation in (10) holds with $M_1 = U$, and $\eta = \theta / \left[ \frac{\theta}{\gamma} + (\theta - 1)(1 - \frac{1}{\gamma} \ln \frac{\theta - 1}{\gamma - 1}) \right]$, which holds based on equation (7).

Since $\beta_1$ is the solution of the first equation in (10) with $M_1 = U$, OTA is $(\gamma, \eta)$-competitive over $[L, U)$ and $[U]$.

With the competitiveness results in above three cases, OTA with the threshold function (8) is $\eta$-consistent and $\gamma$-robust based on Lemma 4.3.

### A.6 Proof of Theorem 5.1

Let $g(p) : [L, U] \to \{0, 1\}$ denote a conversion function of a deterministic online algorithm for 1-max-search, where $g(p) = 1$ (or $g(p) = 0$) represents converting 1 (or 0) dollar under the instance $\mathcal{I}_p$ before the compulsory conversion in the last step. Based on the same arguments as those for the conversion function of one-way trading, the conversion function of 1-max-search satisfies that (i) $g(p)$ is non-decreasing in $[L, U]$ and (ii) $g(U) = 1$.

Let $\mathcal{I}_{\hat{\Phi}}$ denote the first instance, under which an online algorithm for 1-max-search converts 1 dollar, where $\hat{\Phi} = \inf_{\{p \in [L,U]: g(p) = 1\}} p$ is defined as the conversion price. We claim $\hat{\Phi}$ of any $\gamma$-robust online algorithm is upper bounded by $\gamma L$. This claim can be proved by contradiction. Suppose the conversion price of a $\gamma$-robust algorithm is $\gamma L + \varepsilon, \varepsilon > 0$. The profit ratio of the offline optimal and online algorithm under the instance $\mathcal{I}_{\gamma L + \varepsilon/2}$ is $\mathtt{OPT}(\mathcal{I}_{\gamma L + \varepsilon/2})/\mathtt{ALG}(\mathcal{I}_{\gamma L + \varepsilon/2}) = (\gamma L + \varepsilon/2)/L > \gamma$, which contradicts with the $\gamma$-robustness of this algorithm.

Given a prediction $P = U$, to ensure $\eta$-consistency, any $\gamma$-robust online algorithm must have $\eta \geq \mathtt{OPT}(\mathcal{I}_U)/\mathtt{ALG}(\mathcal{I}_U) = U/\hat{\Phi} \geq U/(\gamma L) = \theta/\gamma$, where the second inequality is due to the constraint $\hat{\Phi} \leq \gamma L$ from $\gamma$-robustness.

Based Theorem 4.5, OTA with the reservation price (6) achieves the robustness-consistency trade-off $\eta = \theta/\gamma$, which matches the lower bound, and thus is Pareto-optimal for 1-max-search.

## Appendix B  Detailed Experimental Setup

We use historical Bitcoin (BTC) prices in USD of 5 years from October 2015 through December 2020, with exchange rate information collected every 5 minutes. Our dataset uses publicly available BTC exchange rates gathered from the Gemini cryptocurrency exchange. BTC has gone through dramatic price highs and lows over the years, with a minimum exchange rate of \$353 and maximum exchange rate of \$29,305 by the end of 2020.

In the experiments, each instance captures a trading period of one week and assumes one unit of BTC is available to be traded. Notably, BTC is traded 24/7, so each one week trading period is composed of $2016 = 12 \times 24 \times 7$ five-minute exchange rates. Over the course of 5 years, there are 250 instances of 7 days. In order to facilitate additional rounds in online learning experiments, $\mathtt{Alg}(\lambda^{\mathtt{alf}})$ learns

over 583 instances of 7 days that each overlap by 3 days. If an instance is the period 1/01/2016 to 1/07/2016, the next overlapping instance is 1/04/2016 to 1/10/2016.

To generate a simple prediction $P$ of a one week instance, we use the observed maximum exchange rate of the previous week. With prediction error $\varepsilon = |OPT - P|$, we also test the effect of varying prediction quality by adjusting $\varepsilon$ offline with a multiplicative error level between 0 and 1.0, where 0 error level indicates perfect predictions and 1.0 level indicates unadjusted predictions. To evaluate the performance in worst-case settings, we also introduce a *crash probability* $q$, where the exchange rate of BTC in the last timeslot of the one-week trading period is equal to the lower bound $L$ with probability $q$. In fact, BTC experienced a drop of over $19,000 in a single week of May 2021 following news of Tesla and financial institutions in China no longer accepting BTC as payment.

We report the empirical profit ratio, which is the profit of the optimal offline algorithm over the profit of an online algorithm. This is the counterpart of the theoretical competitive ratio in the empirical setting.