# OpenReview forum: "Pareto-Optimal Learning-Augmented Algorithms for Online Conversion Problems"
_NeurIPS.cc/2021/Conference — NeurIPS 2021 Poster_

### Official Review · Reviewer_q5Tj · 2021-07-16

**Rating:** 7
**Confidence:** 4

**Summary:**

The paper considers learning augmented algorithms for the integral and fractional online conversion problem. One has one unit of some asset that he wishes to convert into another asset. In each step $n = 1,\dots N$  the conversion rate $v_n$ becomes known to the algorithm, and the algorithm decides what amount $x_n$ (possibly $x_n=0$) of the asset should be converted in this step to obtain $x_n v_n$ of the new asset. Summing over all steps, the algorithm obtains in total $\sum_{n\in[N]}x_nv_n$ many units of the new assets, and naturally the objective is to maximise this. In the integer version the whole asset must be converted in only one step, whereas in the fractional version the asset can be converted piece by piece over different steps.

The paper designs a learning augmented online algorithm that is Pareto optimal between robustness and consistency.

Experimental results are presented where the flunctuation of the bitcoin exchange rate over the last five years is used as input data.

**Ethical Concerns:**

no ethical concerns.

**Limitations And Societal Impact:**

no limitations, no social impact I can think of.

**Main Review:**

The algorithm developed in the paper is a so-called online threshold-based algorithm. In particular every time a conversion rate is revealed, the algorithm calculates (using a simple rule) the amount of asset that is to be traded in this step based on the amount that has already been traded, the current conversion rate and a specific threshold function. The challenge lies in defining this threshold function. For the integer version of the problem this function is just a constant thershold (in turn the algorithm will trade on the first step on which the conversion rate is higher than that threshold). For the fractional version of the problem the function is a bit more involved and is built upon the optimal competitive ratio.

The algorithm designed in the paper is elegant and the authors put quite a lot of effort into conveying the intuition behind their ideas. They also provide tight lower bounds implying that their algorithm achieves Pareto optimal tradeoffs between robustness and consistency.

Parsing the experimental results was quite difficult, and despite going over the section several times, I still cannot say what the actual performance of the algorithms is on these experiments. I would recommend to at least give more descriptive names to the variables and to the plot labels and to extend the captions.

Overall, the paper is presenting interesting and -- to the best of my knowledge -- original results. I did not carefully check all the calculations but at least at an intuitive level it all checks out.

Minor comments to the authors:
- your use of $\epsilon$ is a bit confusing. In general $\epsilon$ is used for arbitrarily small values, whereas you use it for the prediction error, which is commonly denoted with $\eta$ and can be quite large in the worst case.
196: neighbour*hood*

**Time Spent Reviewing:**

3

---

> ### Author Response · Authors · 2021-08-10
> **improving the presentation of the experimental results**
>
> Thanks a lot for your positive comments on this paper. Due to the limited space of the submission, we compress the content of the experimental results a lot and thus, the current version may not be clear enough. We will carefully revise the paper to improve the presentation of the experimental results based on your suggestions.
> Regarding your concerns on the notation $\epsilon$, we didn't use $\eta$ to denote the prediction errors mainly because it is already reserved for the notation of consistency. We will consider changing $\epsilon$ to $\varepsilon$ or $\xi$ to avoid the confusion.

---

### Official Review · Reviewer_U9U6 · 2021-07-16

**Rating:** 5
**Confidence:** 4

**Summary:**

The paper considers the following conversion problem: You start with a good (e.g. 1 bitcoin) that you can convert into another good (e.g. dollars) at a conversion rate that changes over time. Every day you have the opportunity to convert part of the good at the day's rate. On the last day, you have to convert any remaining amount of the original good that you might still own. The goal is to maximize the final amount owned. It is assumed that upper and lower bounds U and L on the conversion rate are known that hold throughout the time horizon.

Two versions of this problem are considered: 1-max search, where the good is unsplittable, i.e., one can only convert it as a whole. And a smoother version where arbitrary fractions can be converted, called one-way trading. It seems to me that one-way trading is equivalent to the randomized version of 1-max search.

Let $\theta=U/L$ be the ratio between the upper and lower bounds. In the classical online setting, for 1-max search it is known that the best competitive ratio is $\sqrt{\theta}$, and for one-way trading it is $1+W((\theta-1)/e)$, where W is the Lambert function.

This paper considers this problem through the learning-augmented lens: The algorithm receives as additional input at time 0 a prediction of the maximal conversion rate that will be observed. The paper gives algorithms for both problems and proves via upper and lower bounds that they achieve the pareto-optimal trade-off between consistency (i.e., the competitive ratio when the prediction is correct) and robustness (i.e., the competitive ratio in the worst case). For 1-max trading, the result is that if the consistency is a multiplicative factor c better than the classical competitive ratio $\sqrt{\theta}$, then the robustness is a factor c worse than $\sqrt{\theta}$. The result for one-way trading is more difficult to express and I have no clear intuition of the trade-off. The paper also includes a brief experimental evaluation.

**Ethical Concerns:**

No ethical concerns

**Limitations And Societal Impact:**

Yes.

**Main Review:**

Learning-augmented online algorithms have been a very active research area in recent years, and this paper finds optimal bounds for the conversion problem in this setting, which was not analyzed in this setting before.

The paper shows that some naive algorithms fail, and the constructed algorithms are non-trivial. There is a chance that the techniques in this paper could also be used to derive more fine-grained results for conversion problems, i.e., to establish bounds that show how fast the competitive ratio degrades as the prediction error increases. The authors also say that their lower bound technique could be useful for other problems, though to me it would seem that the main steps for the lower bound is to identify an appropriate family of worst-case inputs (which is a very simple one here), and turning it into a lower bound proof is then usually a matter of calculation (which is non-trivial here though). The authors emphasize that tight lower bounds on the pareto-optimal trade-off of consistency and robustness were  previously known only for the ski rental problem. While true (I think), one reason is also that for many problems it is simply not necessary to make such a trade-off: E.g. for caching, MTS, load balancing etc one can use predictions to improve performance with good predictions without having to make any (or only insignificant) sacrifices in the worst-case performance. This is different for the conversion problem, where bad predictions come at a higher cost.

I am a bit unsure about how convincing I find the overall motivation for the setting of this problem: Since even in the classical online setting it is assumed already that upper and lower bounds U and L on the conversion rate are known, essentially these are already predictions, so in some sense one could say this paper adds a third prediction to a setting where there were already two.

In terms of writing, I found this paper relatively hard to read, and I think much of it could be improved by changing the order in which things are presented. For example, the text at the beginning of section 3.2 is hard to understand for someone that has not read Definition 3.2, Lemma 3.3 and Theorem 3.4 yet. The intuition for the algorithms is also hard to appreciate because it is only given much later than when the algorithms were defined. For example, reading the definition of the six constants in (9) and (10) without any intuition is not particularly enjoyable. Also, no real intuition for Theorem 3.4 is given.

In the experimental setup, some things are unclear to me: How do you choose U and L in the experiments? Are the experiments for the 1-max search algorithm or the one-way trading algorithm? Why not both? Is the adjustment of the error level between 0 and 1 for Figure 3a or also the other figures, and do you actually use 0 somewhere or just 0.6 and 1? For these reasons, I find it hard to evaluate the experimental results.

Overall, I think this is a borderline paper.

Additional comments:
- Lines 50-53: The answer to the question *whether* there exists a pareto-optimal algorithm seems to be trivially Yes. The set of pairs of achievable consistency and robustness is well-defined (even if one does not know the set) so it has a well-defined pareto-frontier. The question is rather *what* those pareto-optimal algorithms are and where the pareto-frontier lies.
- Lines 139-143: The proof of Prop 3.1 only shows an upper bound, but in your discussion you treat it as a lower bound. I agree though that the upper bound in Prop 3.1 seems to be tight.
- Theorem 3.4: The three "if"s in the three cases confused me (where is the "then") and might just be deleted.
- In eq (12) you seem to be doing several steps at once and it would be helpful to break it down.
- Citation [12] seems to be missing the title of the article


COMMENTS AFTER AUTHORS' RESPONSE:

A large part of the response emphasizes the importance of obtaining rigorous lower bounds. I agree and my original review did not say otherwise.

However, the paper is suggesting that the reason why robustness-consistency lower bounds were rarely shown in prior work is due to the difficulty of showing them. While there may exist such problems, I would argue the opposite that the main reason they were not shown/mentioned in most previous works is that often they are simply trivial: E.g. for any MTS (caching and other problems), there is an algorithm with consistency $1+\epsilon$ and robustness $c+\epsilon$, where $c$ is the best competitive ratio without predictions, so the pareto frontier is the single point $(c,1)$ (e.g. for caching, $c=\Theta(\log k)$) and the lower bound follows trivially from the lower bound of the setting without predictions. Unlike for those other problems, where predictions can be incorporated without significant sacrifices in worst-case performance, this is not possible for the conversion problem and thus it is good they provided the lower bound, but I'm not expecting major impact from this on other problems.

The interpretation of L and U as stop-loss and take-profit prices makes sense. In the experiments the authors apparently did use L and U as additional "predictions" though by setting them to the minimal/maximal conversion rates that will be observed, which is information that would not be available to an actual algoritihm.

Overall, the paper still seems rather borderline to me (for NeurIPS standards).

**Time Spent Reviewing:**

6

---

> ### Author Response · Authors · 2021-08-10
> **clarifying the motivation of the problem**
>
> Thank you for your careful reading of our paper. We appreciate your suggestions, especially for the presentation of the paper, and we will revise the paper accordingly. Below we will mainly address your concerns on the motivation for the setting of the problem.
>
> * **(about the lower bound for the robustness-consistency trade-off)**
> First, we would like to clarify the necessity of deriving the lower bound for the learning-augmented online algorithms. From the theoretical aspect, it is always important to rigorously derive the the lower bound in order to quantify the ability of an algorithm to exploit accurate predictions while ensuring robustness to poor predictions. Without a lower bound result, it is not rigorous to conclude *one can use predictions to improve the performance without making any (or only insignificant) sacrifices in the worst-case performance*. In addition, as the reviewer also mentioned, there is no free lunch in the online conversion problems. To achieve a good consistency, the robustness must be sacrificed. Thus, it is well motivated to explore the lower bound for the robustness-consistency trade-off and design algorithms to achieve this lower bound in this paper. Furthermore, we would like to emphasize that there exist additional challenges in deriving the lower bound for the robustness-consistency trade-off, which is also one reason why the lower bounds are not shown in most of prior work. Compared with the single-criteria lower bound in the classical online problems, the learning-augmented problem needs to derive the lower bound for two criteria (i.e., robustness and consistency), and this requires additional techniques to characterize the trade-off between consistency and robustness for any online algorithms even when the worst-case instance is known. The technical novelty of this paper is exactly how to properly characterize the dependence of the robustness and consistency, and further derive the lower bound after figuring out the worst-case instance. To the best of our knowledge, the only known approach to do so is to formulate an optimization problem and exploit the structure of the optimization to derive the lower bound, and this approach has only been successfully applied to the classical ski rental problem (see [4][16]). Thus, our proposed approach can be considered as a promising alternative way of deriving the lower bound, which is of importance even when the worst-case instance is not very difficult to figure out.
>
> * **(about the upper and lower bounds of the conversion rates)**
> The assumption of the bounded conversion rates has been widely used in the literature (including the online conversion problems [6][7] the online knapsack problems [15][17][18], and time series search problem [10]), and it is a necessary assumption in order to derive any meaningful competitive results when the rates arrive in the worst-case order. Thus, we are motivated to follow this assumption to derive theoretical results in this line of research. Additionally, in practical trading problems, $L$ and $U$ can often be considered as predetermined parameters instead of predictions. For example, since the conversion rates in financial markets (like cryptography market) fluctuate dramatically, the investor usually needs to set stop-loss and take-profit prices as part of the exit strategy. Once the real-time rate becomes below the stop-loss price or above the take-profit price, the investor will trade all its remaining asset at the extreme rates to control the losses or lock the profits in this round of trading. Thus, $L$ and $U$ can be naturally considered as the stop-loss price and take-profit price that are predetermined in the trading problem, and they do not conflict with the prediction of the maximum price (which is within $L$ and $U$) that is used to make real-time conversion decisions.
>
> * **(clarifying the settings in the experiments)**
> In experiments, we didn't intend to study the impact of $L$ and $U$, and thus we simply set them as the historical minimum and maximum prices over the entire 5 year period. We only evaluate the one-way trading algorithm since Bitcoin is traded in the unit of satoshi (i.e., $0.00000001$ BTC), which fits the fractional conversion setting. In Figure 3(a), we only evaluate error levels of $0.6$ and $1$, since an error level of 0 indicates the rather unrealistic setting of perfect predictions. We describe the range of the error level between 0 and 1 for clarity of the presentation. We appreciate the reviewer's comments on these points and will clarify them in the revised paper.
>
> * **(about the writing and additional comments)**
> We appreciate the reviewer's detailed suggestions on the writing of the paper, and we will try our best to revise the paper to improve its presentation. Below please find the quick response to the additional comments.
>
>     * We will change the question in line 50-53 to *What is the Pareto-optimal trade-off for the online conversion problem and can we design a Pareto-optimal LOA to achieve it?*
>
>     * The reviewer is correct that the consistency result in Proposition 3.1 is tight. We can see it as follows. When the prediction is accurate and $P = V = \sqrt{LU} - \epsilon$ with $\epsilon\to 0$, the consistency approaches $\sqrt{\theta}$ (see equation (17) in Appendix A.1). We will add additional comments to this point in the revised paper.
>
>     * We missed "then" in the presentation of Theorem 3.4 and will revise it. For example, Case I will be revised to *if $M_{i} \le \phi(0)$, then $M_{i} \le \alpha_{i} L$ and $\beta_{i}=0$*.
>
>     * Due to the space limit, we hide a few steps to get equation (12) and we will show the details in the revised paper.

---

> ### Author Response · Authors · 2021-08-31
> **Response to the additional comments**
>
> Thanks for your additional comments.
>
> * First, as we have mentioned in our last response and the reviewer also agreed, regardless of the importance of trade-offs in other problems, the consistency-robustness trade-off in this online conversion problem (or related competitive search problems) is crucial and non-trivial to achieve. We also would like to clarify that we do not claim our lower bound can directly impact other problem. Instead, we expect our proof techniques can be useful in deriving the lower bounds of other problems.
> In addition to the lower bound, we also expect our proof techniques on the new piece-wise definition of the competitive ratio (Theorem 3.4) can be useful for the competitive analysis of other problems.
>
> * Second, we agree that for some problems, such as the MTS problem [2] mentioned by the reviewer, the trade-off between robustness and consistency cannot be directly observed and seems insignificant when the upper bounds are presented with additional additive terms in the definition of the robustness and consistency (e.g., Theorem 2 in [2] for the randomized algorithm of MTS) or in the big $O$ notation (e.g., Theorem 3 in [2] for the online caching). However, it does not mean there is no trade-off or this trade-off is insignificant in practice. In particular, the reviewer mentioned the $(1+\epsilon)$-consistency and $(1+\epsilon)c$-robustness upper bounds of MTS based on Theorem 2 in [2]. However, such result adopts a different definition of the robustness and consistency by introducing an additive term $O(D/\epsilon)$.
> The trade-off between consistency and robustness is just not observable with the additive term in Theorem 2 and a large additive constant is also not tolerable in practice. To support this point, we notice a recent preprint [*], which studies the online $k$ server problem (a special case of MTS) and shows that the trade-off between the consistency and robustness does exist in Theorem 1 when the definition of the robustness and consistency doesn't include the additive term (in the class of deterministic algorithms) and its proposed algorithm is claimed to outperform the algorithm in [2]. In addition, we can also observe the trade-off for online caching problem if we focus on the absolute value of the robustness and consistency instead of their order forms, see Theorem 4.1 [11].
>
> * Third, we cannot agree that the trade-offs in most previous works are often simply trivial. We would like to argue that this MTS result (Theorem 2 in [2]) is in fact surprisingly good but rare in the literature of learning augmented algorithms. To the best of our knowledge, the majority of the learning augmented algorithms do have non-trivial trade-offs between the consistency and robustness, such as the ski-rental problems [14][16], online set cover [4], secretary and online matching [3], etc. In addition, deriving the lower bound itself has been recognized as an important theoretical contribution in the literature. A key contribution of the paper [16] (Neurips 2020) is to derive the lower bound of the consistency-robustness trade-offs for existing algorithms in [14] of the ski-rental problem. Thus, a novel approach to derive the lower bound (proposed in this paper) can have good potential to impact other problems for deriving (tighter) lower bounds.
>
> [*] A. Lindermayr, N. Megow, and B. Simon, "Double Coverage with Machine-Learned Advice," arXiv:2103.01640 (2021).

---

> > ### Comment · Reviewer_U9U6 · 2021-08-31
> > **Re: Response to the additional comments**
> >
> > It is usually accepted that constant factors play a minor role in problems where O-notation is used. I’m not sure which result of [*] you are referring to. Their lower bound on consistency-robustness for k-server is for a very special class of algorithms (memoryless and other restrictions). For general algorithms, the same MTS result applies. Same for caching, where the MTS result applies. Similarly also for the online set cover problem you mention: Setting $\lambda=1/2$ in [4], one obtains an algorithm with consistency $O(1)$ and robustness $O(\log d)$, which is within a constant factor of the trivial lower bound. Part of the motivation for learning-augmented algorithms is the ability to achieve O(1)-consistency while only losing a constant factor in the worst case. Unfortunately, the conversion problem does not have this property.

---

> > > ### Author Response · Authors · 2021-09-01
> > > **Additional response**
> > >
> > > Thanks for your additional comments and we are now more clear about the conflicting point among us and the reviewer. First, we still would like to clarify that our current discussion is a bit out of the scope of this Neurips paper but we believe this discussion would be good for motivating the emerging filed of learning-augmented algorithms and the study of consistency-robustness trade-offs.
> > >
> > > Our conflicting point is that the reviewer considers that it would be enough to derive **one online algorithm** that can achieve order-optimal consistency and robustness results, i.e., the consistency is $O(1)$ and the robustness is $O(c)$, where $c$ is the optimal competitive ratio.
> > > It seems that the reviewer thinks that if the algorithm is order-optimal, there is no need to further study the trade-offs between consistency and robustness, and the corresponding lower bound. We agree that an order-optimal algorithm is good and we appreciate this type of results. However, what we argue is that the order-optimal result is (i) not good enough, and (ii) may hide the intricate trade-offs of consistency and robustness that cannot be omitted in practice.
> > > In particular, using the big-$O$ notation, exact values can differ by a constant additive term and/or a constant multiplicative term, which make big differences in practice. For example, $1$-consistency and $10$-consistency can both be order optimal, but in practice we aim for an algorithm that tries to achieve $1$-consistency (or close to $1$) at a certain cost of achieving it (measured by robustness) if we believe the prediction is relatively accurate.
> > > Thus, the learning-augmented algorithm is beyond designing a single order-optimal algorithm, and is more interested in designing **a class of parameterized online algorithms** (usually indexed by $\lambda\in[0,1]$ indicating the confidence in the predictions). It is even possible that each online algorithm in this class (for any given $\lambda$) is order-optimal. The main focus is to examine the dependence of consistency and robustness on $\lambda$ and the trade-offs between their exact values.
> > >
> > > * We first use the online set cover problem [4] mentioned by the reviewer to illustrate above point. In Theorem 1 in [4], the consistency is $O({1}/{(1-\lambda)})$ and robustness is $O(\log(d/\lambda))$. For any given $\lambda \in (0,1)$ (as mentioned by the reviewer to set $\lambda = 1/2$), you may claim the consistency and robustness are order-optimal. However, the key contribution of Theorem 1 is to figure out the dependence of the consistency and robustness on $\lambda$ (i.e., the parameteried algorithm) and show the trade-off, i.e., how the robustness evolves as the consistency changes. The paper [4] doesn't show the lower bound for this trade-off but it doesn't mean it is not needed. Moreover, the study of consistency-robustness trade-off provides a class of online algorithms (which are all order-optimal), which gives the design space for selecting the appropriate algorithm that can work best in practice.
> > >
> > > * Second, we take the online caching problem in the early paper [11] as an example (which is also mentioned in our last response). We use this example to show that big-$O$ notations can hide the trade-offs between consistency and robustness.
> > > The competitive ratio of the designed learning-augmented algorithm is $2 \min\\\{1+2S(\epsilon), 2 H(k) \\\}$ (Theorem 3.3 in [11]). To see the trade-offs between the consistency and robustness, the paper further designs a class of online algorithms indexed by $\gamma$ and the competitive ratio is given by $2 \min\\\{1+(1+\gamma)S(\epsilon)/\gamma, \gamma H(k),k \\\}$ (Theorem 4.1 in [11]), which clearly shows the consistency-robustness trade-offs. If we just focus on the results in the big-$O$ notation, such trade-off is not observable.
> > >
> > > * Last, we discuss the MTS problem in [2]. As we have mentioned in our last response, the consistency and robustness of the randomized algorithms (Theorem 2 in [2]) are defined in a different way (with an additive term) and thus the trade-off cannot be observed just based on this result. However, we can compare the result for the deterministic algorithm in Theorem 1 in [2] with that in Theorem 1 in a new preprint [$\ast$]. The MTS result (Theorem 1 in [2]) shows that the competitive ratio of the online $k$-server problem is $9$-consistent and $9 k$-robust. This result can already be considered as being order-optimal, i.e., $O(1)$-consistent and $O(k)$-robust. However, it doesn't mean there is no trade-offs between robustness and consistency. To see the trade-off, Theorem 1 in reference [$\ast$] shows that it can design a class of algorithms parameterized by $\lambda$ that can trade-off the consistency and robustness. And these results outperform those of the MTS result (Theorem 1 in [2]) for some $k$ and $\lambda$. The results in [$\ast$] are within a restricted class of algorithms (with memoryless properties). Then the consistency and robustness (i.e., upper bound) results are still valid. This means there do exist trade-offs between the robustness and consistency in the general MTS problem. We just still cannot figure out tight lower bounds of this trade-off (i.e., lower bound).
> > >
> > > [$\ast$] A. Lindermayr, N. Megow, and B. Simon, "Double Coverage with Machine-Learned Advice," arXiv:2103.01640 (2021).

---

### Official Review · Reviewer_zNZb · 2021-07-16

**Rating:** 6
**Confidence:** 4

**Summary:**

The paper considers learning-augmented algorithms for the online conversion problem. In the online conversion problem, a trader starts with a unit of money in the source currency. Given a sequence of conversion rates v_i that arrive online, the online conversion problem is to determine what fraction of their money to convert using the current rate so that the total amount of money in the target currency is maximized. The paper considers this problem in both the integral and fractional settings. The primary focus of the paper is to design online algorithms for the online conversion problem when provided with a prediction for the maximum conversion rate that achieves the optimal tradeoff between robustness (competitive ratio) and consistency (competitive ratio when prediction is correct).

**Main Review:**

The paper considers the online conversion problem and designs a pareto optimal family of algorithms that utilizes predictions about the maximum conversion rate.

The paper first presents a general family of online threshold algorithms (OTA) for conversion problems. I’m a bit unsure about the novelty of this section since OTA’s have been known for online knapsack and one-way trading (essentially fractional online knapsack) from prior work. Even the general framework described here also appears in [15] almost verbatim (see algorithm 1 there). The paper also presents an interesting notion of “generalized competitive ratio” that gives a more fine-grained view into the worst-case behavior of the algorithm on different types of instances.

Overall, the paper is interesting and designs a non-trivial learning-augmented algorithm for a well-studied online problem. Technically, however, the paper feels a bit lacking and in particular adds limited novelty over prior work in [15], [18], [*]

Strengths: It’s great that the paper also demonstrates tight lower bounds and shows that the proposed algorithm is pareto-optimal.

Other comments:
- The connection with prior work needs to be established better. For instance, the online conversion problem defined here is the fractional version of the online knapsack problem in [18].
- Line 46: “metrical task systems”
- Some discussion of how the predicted value can be learned and why the absolute value of the prediction and the actual max conversion rate is the appropriate notion of error in this context would be useful.
- The references are out-of-date. For instance, [3], [11] have appeared at NeurIPS 2020 and ICML 2018 respectively.

[*] Data-driven Competitive Algorithms for Online Knapsack and Set Cover. Ali Zeynali, Bo Sun,
Mohammad Hajiesmaili, Adam Wierman. AAAI 2021.


**Time Spent Reviewing:**

2

---

> ### Author Response · Authors · 2021-08-10
> **clarifying the technical novelty over prior work**
>
> Thanks for your careful reading of the paper and your good suggestions. We will revise the paper based on your comments. Here we would like to emphasize the technical novelty of this paper over prior work (e.g., [15], [18], [\*]) and briefly discuss the settings of the predicted value and prediction errors.
>
> * **(about the technical novelty over prior work)**
> We would like to clarify that we do not claim the OTA framework as our contribution. As the reviewer mentioned and also what we introduced in the paper (Page 3 line 91-94), OTA has been a well-known framework to solve the online fractional knapsack (one-way trading) [15] or online knapsack with small weights [18][\*]. Our (minor) contribution here is to find that OTA can be used to unify the algorithmic design of both integral and fractional online conversion problems. Previous works have not shown that online algorithm for 1-max-search can be unified into the OTA framework. Based on this OTA framework, we design consistent and robust algorithms using predictions. Compared to prior work using OTA, the key technical novelty of this paper is twofold.
>
>     First, we characterize the general form of the threshold function and the corresponding generalized competitive ratio (Theorem 3.4), which can be used to design Pareto-optimal consistent and robust online algorithms for both integral and fractional conversion problems. Existing works (e.g., online knapsack with small weights [18][\*] or online fractional knapsack [15]) all focus on designing the threshold function of OTA to optimize a single criteria (i.e., competitive ratio). Contrastingly, this paper extends the design of the threshold function to *optimize two criteria (i.e., consistency and robustness) and achieve the Pareto-optimal performance*. This requires a non-trivial extension of prior work and we believe it is a main technical contribution. With regard to the ref [\*] mentioned by reviewer, [\*] considers designing a family of online algorithms by parameterizing the threshold function and guaranteeing a bounded competitive ratio for each algorithm in this family. However, the algorithms in [\*] take no advantage of the predictions and can still only guarantee a single criteria. The consistent and robust algorithm designed in this paper is also a family of online algorithms but each algorithm achieves the Pareto-optimality of two criteria and is guaranteed to achieve good performance with good predictions.
>
>     Second, to derive the lower bound for the robustness-consistency trade-off, this paper also overcomes additional technical challenges compared to prior work (e.g., [15][18]). In particular, it is usually sufficient to derive the lower bound for a single-criteria online problem if we can figure out its worst-case instance. However, in this paper, we need to derive a lower bound with two criteria. In addition to finding the worst-case instance, we also have to find a way to characterize the trade-off of the two criteria. In the literature, the only known tight lower bound for the robustness-consistency trade-off is derived for the ski rental problem (in refs [4][16]) and their approach is to formulate an optimization problem to capture the trade-off, and further derive the lower bound based on the structure of the optimization problem. Our paper provides an alternative approach to derive the lower bound by constructing a function that can bridge the requirements of robustness and consistency for any online algorithms. We believe this constructive approach to achieve the lower bound is novel compared to prior work.
>
>
> * **(about the predicted value and prediction errors)**
> In the online conversion problem, if the actual maximum rate of the trading period is known, we just need to trade all assets at this rate and achieve the maximum profit.
> Thus, the maximum rate is an appropriate value to be predicted and incorporated into the design of the online algorithm.
> In the learning-augmented online algorithm, the predicted value can be obtained from any blackbox predictor (possibly based on machine learning). The design and analysis of our algorithm are independent of how the predicted value is obtained but can guarantee a good performance if the prediction is accurate. Since the prediction is just a scalar, the absolute difference between the prediction and the actual value is actually a natural choice. We will add more discussions about the predicted value and prediction errors to make above points clear in the revised paper.

---

### Official Review · Reviewer_BJtR · 2021-07-19

**Rating:** 6
**Confidence:** 4

**Summary:**

This paper provides learning-augmented algorithms for online conversion problems. Suppose you have $1 and want to convert it to a different currency where the exchange rate varies over time. The ideal would be to do the conversion when the exchange is at its highest, but at any time, we do not know if the current exchange rate is in fact the highest (similar to secretary problems). The paper considers both the integer and fractional versions of this problem, i.e., when the conversion of the entire one unit has to be done all at once or when it can be done in fractions multiple times adding up to one. Algorithms are presented for these problems if a prediction of the maximum value is given in advance, the goal being to achieve the optimal tradeoff between consistency and robustness. Matching lower bounds on this tradeoff are also presented. Finally, an empirical evaluation of the algorithms are carried out for the application of trading dollars for bitcoins.

**Ethical Concerns:**

None that I can think of.

**Limitations And Societal Impact:**

No potential negative societal impact that I can think of.

The authors provide some limitations of their results, particularly in terms of not proving any results that establish graceful degradation, and not allowing the robustness parameter to be chosen online. These are indeed limitations, but the shortcomings mentioned above in terms of lack of technical novelty are more glaring.

**Main Review:**

There are three main contributions of the technical parts in the paper. The first is a general characterization of the threshold function in the online algorithm, and the resulting tradeoff between robustness and consistency (Theorem 3.4). [Lemma 3.3 can be considered to be a restatement of the definitions of consistency and robustness.] The second contribution is to apply this general characterization to the two conversion problems: integral and fractional, and derive corresponding online threshold-based algorithms. The third is to use the characterization to achieve lower bounds on the tradeoff. Of these, the second and third contributions are actually pretty easy if considered directly (without the characterization). For instance, the algorithm for setting a threshold that achieves $\eta$ consistency and $\gamma$ robustness in the integral conversion case is based on the following simple observations: (a) do not set a threshold about $U / \eta = L \gamma$ since it is sufficient to ensure $\eta$ consistency for $P > U / \eta$, (b) do not set a threshold below $L \gamma$ since it is enough to ensure $\gamma$ robustness by selling at the final price even if that is just $L$, (c) between these extremes, use a convex combination of $L \gamma$ and $P / \eta$ to trade off between consistency and robustness. The fractional case is a little more complicated because the threshold is now a function, but it also follows by applying a similar logic and using the existing online algorithm for this problem (without prediction). Similarly, the lower bounds can be derived by creating instances that represent the limiting cases of the observations given above. So, it is not clear if the complicated framework established by Theorem 3.4 is required for the subsequent theorems for the actual problems. In addition to this lack of technical novelty and clarity of purpose, the problem itself is only of modest importance, and one that is closely related to other problems like secretary problems and prophet inequalities that have been previously considered in the prediction model. (In fact, the authors should add a clear comparison of their results to the existing ones for these problems given the close resemblance.)

The response to the authors to my initial review was adequate and addressed some of the many shortcomings that I had outlined. I have increased my score accordingly.

**Time Spent Reviewing:**

1

---

> ### Author Response · Authors · 2021-08-10
> **clarifying the technical contributions and the importance of the problem**
>
> Thanks for your comments. Below we clarify the following points: (i) the reviewer's observations are useful but cannot be directly used to design the threshold function. Theorem 3.4 is necessary to design the consistent and robust algorithms for the online conversion problems; (ii) to derive a tight lower bound for the robustness-consistency trade-off, it is insufficient to just figure out the worst-case instance, and our approach to derive the lower bound is novel; (iii) the online conversion problem is indeed of importance from both theoretical and practical aspects, and our contribution in this paper is salient compared to the related works (e.g., secretary problem with predictions).
>
> * **(about the limitations of the reviewer's observations)**
> The reviewer mentions three observations that can be used to design the threshold function (6) directly. First, we clarify that these observations (although some of them are useful) are only partially correct, and cannot be used to design the threshold function (6) directly. The characterization of the threshold function in Theorem 3.4 is still the key to design consistent and robust algorithms for both integral and fractional online conversion problems.
>
>     In more detail, the observation (a) is correct but a more critical reason is missing. When the prediction is $P > U/\eta = L\gamma$, we should not set a threshold $\Phi$ above $U/\eta$ because (i) it is sufficient to ensure $\eta$-consistency by just setting $\Phi = U/\eta$; and *more importantly* (ii) this will result in the worst-case competitive ratio (i.e., robustness) larger than $\gamma$ (when the actual maximum rate is $\Phi-\epsilon$ and $\Phi > U/\eta$, $\text{CR}(\infty) = (\Phi-\epsilon)/L > \gamma$). Similarly, we can also observe that when $P < L\eta = U/\gamma$, we should not set a threshold below $L\eta$ since (i) it is sufficient to ensure $\eta$-consistency by setting $\Phi = L\eta$; and (ii) it will lead to a competitive ratio larger than $\gamma$ ($\text{CR}(\infty) = U/\Phi > \gamma$). A more accurate description of above two observations is that when the prediction is close to boundaries, i.e., $P > U/\eta$ or $P < L \eta$, consistency is easily ensured and we must carefully design the threshold to guarantee the robustness. However, this observation is not sufficient to tell how to balance the consistency and robustness. Based on Theorem 3.4, we can show the robustness of the algorithm is $\max\\\{\Phi/L,U/\Phi\\\}$. Together with above observation, we can finally design the threshold (6a) and (6c). In fact, above discussion is exactly what we explain below Theorem 3.5 (Page 6, line 227-230) in the paper.
>
>     The observation (b) is partially correct. We should not set a threshold below $L\gamma$ *only when the prediction $P > L\gamma$*. However, when $P < L\gamma$, although a threshold $\Phi = L\gamma$ can ensure $\gamma$-robustness, we still must design the threshold to be smaller than $L\gamma$ to ensure the $\eta$-consistency. For example, if the threshold is above $L\gamma$ and the prediction is $P=L\gamma -\epsilon$, the consistency is $\eta = (L\gamma-\epsilon)/L$, which approaches the robustness and is undesirable. To design robust and consistent online algorithms, we must design the threshold function that can work for any instances and predictions. Thus, observation (b) cannot be used to directly design the threshold function. Instead, observation (b) is more useful when we derive the lower bound since we only need to consider a special family of instances and predictions. In fact, we already use this observation in the proof of Theorem 4.2 (Page 8, line 312-316).
>
>     Finally, the observation (c) is not the correct intuition for designing (6b). Since $P \le U$, we have $P/\eta \le U/\eta = L\gamma$. Thus, a convex combination of $L\gamma$ and $P/\eta$ is smaller than $L\gamma$, which already contradicts with the observation (b). More importantly, a naive convex combination of the prediction and some reference value does not necessarily trade-off the robustness and consistency as shown in Proposition 3.1. The threshold function (6b) is indeed a carefully-crafted one. The key of designing (6b) is to characterize the consistency of the online algorithm with threshold $\Phi$ as $\max\\\{\Phi/L,P/\Phi\\\}$ (based on Theorem 3.4). Thus, to ensure $\eta$-consistency, we must enforce $P\ge \Phi$ and the threshold (6b) satisfies this and ensures $\eta$-consistency. We have already provided this intuition in the paper (Page 6, line 232-234).
>
> * **(about the technical novelty of Theorem 3.4)**
> We agree that the characterization of the robustness and consistency for 1-max-search is not very difficult. However, when considering the one-way trading problem, it is much harder to characterize the robustness and consistency without Theorem 3.4. Thus, Theorem 3.4 is necessarily needed to design the threshold function (8) for one-way trading. The key technical novelty of Theorem 3.4 is to provide the general form of the piece-wise threshold function, and the dependence of the multiple competitive ratios and threshold pieces (e.g., in equation (3), the differential equation for the $i$-th piece of the threshold function depends on all previous pieces). This characterization in Theorem 3.4 is essential to derive the threshold function for one-way trading with robustness and consistency guarantees. In fact, each threshold piece in equation (8) is a solution of a set of differential equations (see Appendix A.5) based on Theorem 3.4, and the parameters in each piece are dependent and need to solved by a set of equations simultaneously (i.e., equations (9) and (10)). Without Theorem 3.4, we are not aware how to design this threshold function just based on simple observations and existing online algorithms without prediction. Last, Theorem 3.4 can provide the characterization for both integral and fractional problems. we do think it is a novelty to have a unified technical theorem to guide the design of both problems and provide rigours analysis, instead of focusing on them independently.
>
> * **(about the technical novelty of the lower bound)**
> First, we emphasize the additional challenge of deriving the tight lower bound for the consistency-robustness trade-off compared to the lower bounds for the scalar competitive ratios in the literature of online algorithms. The key difficulty of deriving the lower bound of a single criteria (i.e., competitive ratio) lies in finding the worst-case instance. However, to derive the tight lower bound for multiple criteria (i.e., robustness and consistency), it is not enough to just find the worst-case instance. A more challenging task is to find a proper approach to characterize the dependence of the robustness and consistency for any online algorithm under the worst-case instance. To the best of our knowledge, the only known tight lower bound for consistency-robustness trade-off is for the classical ski rental problem (see refs [4][16]), whose worst-case instance is well known. The key step is to formulate an optimization problem to characterize the dependence between robustness and consistency and derive the lower bound based on the structure of the optimization problem.
>
>     In the online conversion problem, we agree that its worst-case instance is not very difficult to figure out. However, although we tried, we still cannot derive the lower bound using the optimization-based method since the optimization formulation and structure are problem-specific. Thus, we turn to proposing an alternative method for deriving the lower bound. The novelty of our approach is to construct a function that can model all online algorithms under the worst-case instance and derive the lower bound based on the requirements of the robustness and consistency on this function. Although this approach still may not be applied to all online problems, it is a promising candidate in the toolbox of deriving the lower bound.
>
> * **(about the importance of the online conversion problem)**
> The online conversion problem is an important and classical online problem, and its variants are closely related to the online knapsack problems (online $0/1$ knapsack with small weights [18] or online fractional knapsack [15]), and time series search problem [10], etc.). Those problems all aim to search for the maximum from a sequence of rates under adversarial input model (i.e., the rates arrive in the worst-case order and are bounded within $[L,U]$). To the best of our knowledge, this paper is the first one to consider incorporating the untrusted predictions and designing robust and consistent online algorithms in this line of research.
>
>     Secretary problem and prophet inequalities also aim to search for the maximal value but they make additional assumptions on the input. For example, the secretary problem assumes the rates arrive in a uniformly random order and the prophet inequality assumes the rates arrive in the worst-case order but with prior distributions. The additional (statistical) information in these problems makes the design and analysis of online algorithms essentially different from ours. Thus, although there exist results for those problems with predictions (e.g., the secretary [3]), it is unclear how to compare their results with ours. In addition, the online conversion problem under adversarial input model is indeed important in real-world applications such as the Bitcoin conversion, where the conversion rates fluctuate dramatically and cannot be modeled by the random order model or with prior distributions. Thus, we believe this paper is studying a fundamental problem of both theoretical and practical importance.
>
>     We appreciate the reviewer's suggestions on adding discussions about the connections between this work and other classical problems (e.g., the secretary problem) and will include it in the revised paper.

---

### Decision · Program_Chairs · 2021-09-27

**Decision:**

Accept (Poster)

**Comment:**

This paper provides algorithms augmented machine-learned predictions for online conversion problems. The goal is to improve the competitive ratio when predictions are accurate ("consistency"), while also guaranteeing a worst-case competitive ratio regardless of the prediction quality ("robustness"). The authors show their algorithms achieve the Pareto-optimal trade-off of consistency and robustness.

Most reviewers found the paper interesting, original, and having sufficient contribution for acceptance to the conference. They also appreciated elegant design of the algorithms and their formal analysis, and tight lower bounds on the Pareto-optimal trade-off. One of the reviewers felt the paper brings a limited novelty over prior work.

Most of the reviewers raised issues related to the clarity of the presentation. I hope the authors will take into account all these comments to improve the final version of the manuscript.